# You Only Spectralize Once: Taking a Spectral Detour to Accelerate Graph Neural Network

**Yi Li[α], Zhichun Guo[β], Guanpeng Li[γ], and Bingzhe Li[α]**
[α] University of Texas at Dallas
[β]University of Washington
[γ]University of Florida
{Yi.Li3 ,bingzhe.li}@utdallas.edu
zcguo@uw.edu
liguanpeng@ufl.edu

## Abstract

Training Graph Neural Networks (GNNs) often relies on repeated, irregular, and expensive message-passing operations over all nodes (e.g., $N$), leading to high computational overhead. To alleviate this inefficiency, we revisit the GNNs training from the spectral perspective. Node features and embeddings in many real-world graph exhibit sparse representation in Graph Fourier domain. This inherent sparsity aligns well with the Compressed Sensing principles, which posits that sparse signals can be accurately reconstructed from significantly fewer measurements (e.g., $M$ and $M \ll N$). This observation motivates designing efficient GNNs that operates predominantly in a compressed spectral subspace. In this paper, we propose You Only Spectralize Once (YOSO), a GNN training scheme that first performing a single projection of features onto a learnable orthonormal Graph Fourier basis $U_\ell$, and after compressed sensing is used, retaining only $M$ spectral coefficients where $M \ll N$. The entire GNN computation then performs in this reduced/compressed spectral domain. Finally, the full graph embeddings are recovered back to original domain at output layer by solving a compressed sensing bounded $\ell_{2,1}$-regularized optimization problem. Theoretically, drawing upon the compressed sensing theory, we prove that stable recovery by showing that this whole process can satisfy the Restricted Isometry Property when $M = \mathcal{O}(k(\log N/k))$. Empirically, YOSO achieves an average 74% training time reduction across five benchmark datasets compared to state-of-the-art baseline schemes, while maintaining the competitive model accuracy.

## 1 Introduction

Graphs effectively capture the relational structures in diverse data [27, 51, 26, 39], offering advantages over Euclidean representations [7]. Graph Neural Networks (GNNs) [36, 28, 72, 15] leverage this via localized message passing (LMP) [19], achieving state-of-the-art (SOTA) performance in tasks such as link prediction [25, 67] and node classification [71, 69]. However, LMP on the large-scale graphs leads to prohibitively long training time, e.g., billion-scale graphs can demand hours or even days of computation [38, 58]. To mitigate these costs, prior works explored the strategies like graph pruning/condensation [50, 33, 60] and various sampling techniques [13, 31, 66, 15]. By discarding parts of the graph or limiting nodes neighbors exploration, these methods might miss crucial relational information, thus often leading to a notable degradation in model accuracy [12]. Consequently, GNN training faces a fundamental efficiency-accuracy trade-off.

39th Conference on Neural Information Processing Systems (NeurIPS 2025).

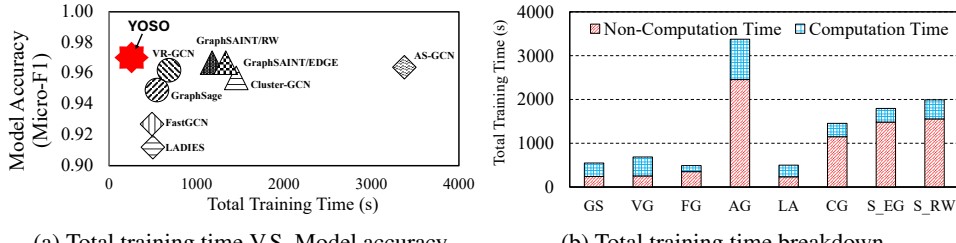

(a) Total training time V.S. Model accuracy          (b) Total training time breakdown

Figure 1: Total training time and model accuracy of different schemes, including GraphSage (GS) [28], VR-GCN (VG) [12], FastGCN (FG) [13], AS-GCN (AG) [31], LADIES (LA) [72], Cluster-GCN (CG) [15] and two different versions of GraphSAINT (S_EG and S_RW) [66], on Reddit [28]. Total training time is broken into two parts: (1) Non-computation time, includes mini-batch construction (e.g., indexing, sampling) and data transfer between host and device memory; and (2) Computation time, which covers computation on the GPU (i.e., forward and backward propagation, and weight updates). The seven-pointed red star marks the desired operational point targeted by this paper.

Fig. 1a illustrates this trade-off: methods such as GraphSAINT [66] and AS-GCN [31] achieve high accuracy with comprehensive neighborhood data but suffer long training times (top-right of Fig. 1a). Conversely, schemes like FastGCN [13] and LADIES [72] offer speed-ups but sacrifice accuracy (bottom-left of Fig. 1a) due to the information loss from aggressive sampling [55]. These existing methods form a Pareto frontier: reducing non-computational overheads (e.g., host-side sampling, host-device data migration etc.) impairs accuracy, while preserving accuracy inflates these overheads. We performed a breakdown analysis, as illustrated in Fig. 1b. The results reveal that non-computation time dominate the overall training time relative to the computation time (i.e., computation in devices), highlighting a key opportunity for optimization.

Our goal is to transcend this trade-off, achieving both high accuracy and low training time (reaching *seven-pointed red star* indicated in Fig. 1a). To this end, we draw inspiration from the Graph Fourier analysis [52, 44, 37, 17] where node features are viewed as signals defined on graph structure [45]. Orthonormal eigenvectors of graph Laplacian [42, 21, 24, 36] (e.g., $L_{sym}$) serve as the fundamental frequency components [35, 61]. Graph Fourier transform matrix $U$ (composed of these eigenvectors) projects original features $X$ into spectral domain via $U^\top X$ [46]. This initial transformation is, in principle, information-preserving [44, 42]. A crucial observation is that graph signals (both initial features and intermediate GNN embeddings) exhibit spectral sparsity [32, 48]. This means that the most important information in these graph signals can be captured by just a few dominant frequencies or spectral components [42], while many others carry little significant information. Retaining only these critical coefficients is a principle central to Compressed Sensing (CS). CS posits that if a signal is sparse in some domain, we don't need to measure all its components to reconstruct it accurately; a much smaller number of "smart" measurements can suffice to recover the essential information. By operating only on these few critical spectral coefficients, we could drastically reduce the amount of data the GNN needs to process, store, and move, directly tackling the non-computational bottlenecks (Fig. 1b) and potentially speeding up the GNN's computations themselves.

However, directly integrating Graph Fourier and compressed sensing principles into GNN training presents several significant challenges: **(a) Prohibitive $U$ Construction Cost:** To use Graph Fourier, it is typically need a fixed $U$, which involves eigendecomposition of graph Laplacian (e.g., $L_{sym}$) that typically has complexity of $\mathcal{O}(N^3)$ [53, 5], where $N$ is total number of nodes in the graph. This makes pre-computing suitable $U$ a substantial overhead itself for large graphs [32, 6, 57, 48, 14]. **(b) Inaccurate Reconstruction from the Spectral Domain:** GNNs iteratively apply the non-linear activation functions. If transform embeddings into spectral domain and later convert them back to original domain (for performing the downstream task), but only use a simple inverse transform (e.g., $U^{-1}$), thus, these non-linearities prevent perfect recovery of original embeddings. This information loss can be accumulated layer-by-layer and degrade final model accuracy [5, 2]. **(c) Costly Layer-wise Sensing and Recovery:** While compressed sensing [10, 9] can, in principle, help to achieve more accurate reconstruction from fewer measurements and potentially counteract some effects of non-linearities [63, 62], applying the full sequence of Graph Fourier transform, compressed sensing measurement and recovery at every GNN layer to maintain embedding fidelity would introduce its own significant computational overhead, negating any time benefits we try to introduce. To address these challenges, we propose the YOSO (Y̱ou O̱nly S̱pectralize O̱nce).

To overcome challenge of fixed basis $\boldsymbol{U}$, YOSO does not pre-compute it but using a learnable $\boldsymbol{U}_\ell$, which is learned as part of the GNN training process via backpropagation. Crucially, this potentially complex transformation is performed only once on initial input features. For compressed sensing (CS) to be effective, data needs to be sparse in chosen transform domain, YOSO's unified training objective includes the terms (in total loss $\mathcal{L}_{\text{total}}$) that encourage $\boldsymbol{U}_\ell$ to naturally induce sparse representations from the input data, denoted $\boldsymbol{X} = \boldsymbol{H}^{(0)} \in \mathbb{R}^{N \times d}$ where $d$ is the dimension of feature vector. After this one-time Graph Fourier transformation $\boldsymbol{U}_\ell^\top \boldsymbol{X}$ and the selection of only $M$ (where $M \ll N$) significant spectral coefficients, all subsequent GNN computations proceed entirely in this highly compact, $M$-dimensional, CS-informed spectral domain. To tackle inaccurate reconstruction and costly layer-wise sensing, YOSO applies CS principles for the final output reconstruction only. It employs a fixed sensing matrix $\boldsymbol{\Phi}$ (designed to work effectively with the learned $\boldsymbol{U}_\ell$ and satisfy a key condition called the Restricted Isometry Property (RIP) [10], which ensures stable recovery from few measurements). Then, a CS recovery algorithm (bounded $\ell_{2,1}$-regularization) reconstructs the full-graph node embeddings from the GNN's compact $M$-dimensional output. This one-time spectral projection at the input and one-time CS-based reconstruction at the output layer significantly reduces computational complexity. Our main contributions are:

- We propose YOSO, a GNN training scheme that leverages a once-per-training learnable orthonormal spectral transformation to operate in a Compressed Sensing–informed compressed spectral domain, thereby mitigating expensive computations (e.g., full message passing), circumventing complex spatial sampling, and significantly reducing training time while maintaining competitive accuracy.

- Experiments show YOSO reduces training time by $\sim 74\%$ on average, preserving accuracy comparable to SOTA methods, with robust performance and high-fidelity embedding reconstruction.

## 2 Background and Preliminaries

**Graph Neural Networks.** GNNs operate on graph $G = (V, E, \boldsymbol{A}, \boldsymbol{X})$, where $V = \{v_1, \ldots, v_N\}$ is the node set, $E \subseteq V \times V$ the edge set, $\boldsymbol{A} \in \mathbb{R}^{N \times N}$ the adjacency matrix, and $\boldsymbol{X} \in \mathbb{R}^{N \times d}$ is node feature matrix. GNNs learn node embeddings $\boldsymbol{H}^{(l)} \in \mathbb{R}^{N \times d^{(l)}}$ for layer $l$ parameterized by $\theta^{(l)}$ via: $\boldsymbol{H}^{(l)} = f_{\theta^{(l)}}(\boldsymbol{H}^{(l-1)}, \boldsymbol{A})$, for layer $l = 1, \ldots, L$. We denote $\boldsymbol{H}^{(0)} \equiv \boldsymbol{X}$.

**Graph Fourier (GF) Analysis.** One important concept in GF Analysis is GF transform, which often uses graph symmetric normalized Laplacian $\boldsymbol{L}_{\text{sym}} = \boldsymbol{I} - \boldsymbol{D}^{-\frac{1}{2}} \boldsymbol{A} \boldsymbol{D}^{-\frac{1}{2}}$ [57, 36, 14], where $\boldsymbol{D}$ is the diagonal degree matrix. $\boldsymbol{L}_{\text{sym}}$ has eigendecomposition $\boldsymbol{L}_{\text{sym}} = \boldsymbol{U} \boldsymbol{\Lambda} \boldsymbol{U}^\top$, where $\boldsymbol{U} \in \mathbb{R}^{N \times N}$ is an orthonormal matrix of eigenvectors (GF basis) and $\boldsymbol{\Lambda} = \text{diag}(\lambda_0, \ldots, \lambda_{N-1})$ contains corresponding non-negative eigenvalues (graph frequencies). The GF transform of $\boldsymbol{H}^{(l)}$ is $\hat{\boldsymbol{H}}^{(l)} = (\boldsymbol{U}^{(l)})^\top \boldsymbol{H}^{(l)}$, with the inverse $\boldsymbol{H}^{(l)} = \boldsymbol{U}^{(l)} \hat{\boldsymbol{H}}^{(l)}$. $\hat{\boldsymbol{H}}^{(l)} \in \mathbb{R}^{N \times d^{(l)}}$ (called spectral coefficients) is sparse for real-world graphs [2, 14, 17, 29, 32], meaning that most energy or information is store in few coefficients, enabling compressed sensing-based reconstruction.

**Compressed Sensing (CS)** can reconstruct signals with few measurements, based on two prerequisites [10]. First, the reconstructed signals need to exhibit sparsity in the transform domain [9], and Graph Fourier analysis provides guarantee for this: $k$-row-sparse $\hat{\boldsymbol{H}}^{(l)}$ in transform domain. The second is Restricted Isometry Property (RIP). CS acquires a compressed measurements $\boldsymbol{T}^{(l)} \in \mathbb{R}^{M \times d^{(l)}}$ using the sensing matrix $\boldsymbol{\Phi}^{(l)} \in \mathbb{R}^{M \times N}$ ($M \ll N$) such that $\boldsymbol{T}^{(l)} = \boldsymbol{\Phi}^{(l)} \boldsymbol{H}^{(l)} = \boldsymbol{\Phi}^{(l)} \boldsymbol{U}^{(l)} \hat{\boldsymbol{H}}^{(l)}$. Operating on the measurements $\boldsymbol{T}^{(l)} \in \mathbb{R}^{M \times d^{(l)}}$ instead of $\boldsymbol{H}^{(l)} \in \mathbb{R}^{N \times d^{(l)}}$ is more efficient since $M \ll N$ [54, 29]. A successful recovery of $\hat{\boldsymbol{H}}^{(l)}$ from $\boldsymbol{T}^{(l)}$ relies on the matrix $\boldsymbol{\Psi}^{(l)}$, defined as $\boldsymbol{\Phi}^{(l)} \boldsymbol{U}^{(l)}$. $\boldsymbol{\Psi}^{(l)}$ should satisfy the Restricted Isometry Property (RIP) [10] with a constant $\delta_k \in (0, 1)$ for all $k$-row-sparse $\hat{\boldsymbol{H}}^{(l)} \in \mathbb{R}^{N \times d^{(l)}}$: $(1 - \delta_k)\|\hat{\boldsymbol{H}}^{(l)}\|_F^2 \le \|\boldsymbol{\Psi}^{(l)} \hat{\boldsymbol{H}}^{(l)}\|_F^2 \le (1 + \delta_k)\|\hat{\boldsymbol{H}}^{(l)}\|_F^2$. If RIP holds, $\hat{\boldsymbol{H}}^{(l)}$ can be estimated from measurements $\boldsymbol{T}^{(l)}$. This estimation, denoted $\hat{\boldsymbol{H}}_c^{(l)*}$, is found by an optimization problem that promotes row-sparsity, commonly via $\ell_{2,1}$-minimization:

$$\hat{\boldsymbol{H}}_c^{(l)*} \equiv \text{Solver}(\boldsymbol{T}^{(l)}, \boldsymbol{\Phi}, \boldsymbol{U}^{(l)}, \lambda) \triangleq \text{argmin}_{\hat{\boldsymbol{H}}^{(l)}} \left( \frac{1}{2} \left\| \boldsymbol{T}^{(l)} - \boldsymbol{\Phi}^{(l)} \boldsymbol{U}^{(l)} \hat{\boldsymbol{H}}^{(l)} \right\|_F^2 + \lambda \left\| \hat{\boldsymbol{H}}^{(l)} \right\|_{2,1} \right) \tag{1}$$

Solver$(\cdot)$ represents a determinative algorithm like FISTA [4] adapted for the $\ell_{2,1}$-norm, and $\lambda$ is a regularization parameter. The asterisk in $\hat{\boldsymbol{H}}_c^{(l)*}$ denotes this is an estimated approximation of the true sparse coefficients. The $\ell_{2,1}$-norm is $\|\hat{\boldsymbol{H}}^{(l)}\|_{2,1} = \sum_{i=1}^N \|\hat{\boldsymbol{h}}^{i(l)}\|_2$ (where $\hat{\boldsymbol{h}}^{i(l)}$ is the $i$-th row

of $\hat{\boldsymbol{H}}^{(l)}$). Once the specific $\hat{\boldsymbol{H}}_c^{(l)*}$ value is solved, then the original domain signal can be computed and reconstructed as $\boldsymbol{H}_c^{(l)*} = \boldsymbol{U}\hat{\boldsymbol{H}}_c^{(l)*}$. Crucially, the number of measurements $M$ required for successful recovery is primarily dictated by the sparsity level $k$ (e.g., $M = \mathcal{O}(klog(N/k))$), not the number of nodes $N$ (see Appendix B.2 for the detailed statement).

## 3 Benefits and Obstacles of Integrating GF and CS in GNNs

Integrating Graph Fourier transform and compressed sensing layer-wise in GNNs seems promising, since training process can be performed with dimension of $M(\ll N)$ instead of original $N$. If node embeddings $\boldsymbol{H}^{(l)} \in \mathbb{R}^{N \times d^{(l)}}$ at each GNN layer $l$ could be represented sparsely via a $\boldsymbol{U}^{(l)} \in \mathbb{R}^{N \times N}$ as $\boldsymbol{H}^{(l)} = \boldsymbol{U}^{(l)}\hat{\boldsymbol{H}}^{(l)}$, one could use a sensing matrix $\boldsymbol{\Phi}^{(l)}$ to compress: $\boldsymbol{T}^{(l)} = \boldsymbol{\Phi}^{(l)}\boldsymbol{H}^{(l)} \in \mathbb{R}^{M \times d^{(l)}}$. One potential successful recovery would involve solving Eq.(1) and $\boldsymbol{\Psi}^{(l)} = \boldsymbol{\Phi}^{(l)}\boldsymbol{U}^{(l)}$ is required to satisfy Restricted Isometry Property (RIP). This offers two theoretical benefits:

**Benefit I: Reduced Data Representation.** Compressed $\boldsymbol{H}^{(l)} \in \mathbb{R}^{N \times d^{(l)}}$ to $\boldsymbol{T}^{(l)} \in \mathbb{R}^{M \times d^{(l)}}$ where $M \ll N$, could reduce computational/memory needs, if subsequent GNN computation could be adapted to work directly with $\boldsymbol{T}^{(l)}$, e.g., use $\boldsymbol{T}^{(l)}$ as GNN input to perform the forward propagation.
**Benefit II: Faithful Signal Recovery.** Compressed sensing provides the guarantee via Restricted Isometry Property, such that $\hat{\boldsymbol{H}}^{(l)} \in \mathbb{R}^{N \times d^{(l)}}$ can be accurately recovered from $\boldsymbol{T}^{(l)}$. This suggests that the full expressive power of the original embedding space could be restored for downstream tasks, for example, node classification and link prediction.

However, integrating Graph Fourier and compressing sensing with layer-wise GNN (superscript $(l)$), aligning with the above description (ideal scenario in Eq.(2)) is impractical due to two obstacles:

$$\boldsymbol{H}^{(l)} = f_{\theta^{(l)}}(\text{Rec}\{\boldsymbol{T}^{(l-1)}\}, \boldsymbol{A}) \tag{2}$$

**Obstacle I: Prohibitive Costs and Complexity of Layer-Specific Transformations.** A layer-wise application of Graph Fourier and compressed sensing would necessitate a distinct orthonormal basis $\boldsymbol{U}^{(l)}$ and a corresponding sensing matrix $\boldsymbol{\Phi}^{(l)}$ for each GNN layer $l$. Determining the optimal $\boldsymbol{U}^{(l)}$ typically requires costly eigendecompositions ($\mathcal{O}(N^3)$ complexity [53, 5]), as a single fixed basis is unlikely to maintain the sparsity for embeddings across all non-linear GNN layers. Furthermore, designing and verifying $L$ distinct $\boldsymbol{\Phi}^{(l)}$ matrices to pair with each $\boldsymbol{U}^{(l)}$ and satisfy the Restricted Isometry Property adds significant complexity, rendering this layer-by-layer approach impractical. This motivates a strategy using a single, efficiently obtained (e.g., learnable) transformation and a universal sensing matrix, denoted $\boldsymbol{\Phi}$ (without superscript $l$).
**Obstacle II: Impracticality of Layer-Wise Reconstruction and the Resulting Efficiency-Accuracy Dilemma.** Even layer-specific transformations were feasible, reconstructing full $N$-dimensional embeddings from compressed measurements at each of the $L$ total GNN layers is computationally prohibitive. Solving the required optimization problem (Eq.(2)) repeatedly (e.g., with complexity related to $\mathcal{O}(NMd^{(l)})$ per layer [41]) would negate any efficiency gains from compressed format.

Therefore, a successful integration should reap the benefits of compression without paying the penalty of expensive reconstructions at every layer. This necessitates a framework that carefully balances the depth of spectral processing against the fidelity of information required for high GNN performance, ideally through a single and efficient transformation and reconstruction pipeline.

## 4 Methodology

We present the design of YOSO in this section. Section 4.1 introduces YOSO's overall architecture and computational workflow. Section 4.2 specifically describe how YOSO learns $\boldsymbol{U}_\ell$ (addressing Obstacle I) and performs GNN computations (addressing Obstacle II). Finally, the universal sensing matrix $\boldsymbol{\Phi}$ (further addressing Obstacle I) details in Section 4.3

### 4.1 Overall Process of YOSO

As illustrated in Fig. 2, YOSO introduces a novel GNN training pipeline that integrates a learnable transformation $\boldsymbol{U}_\ell$ with compressed sensing principles. The core idea is that to project features into a learnable, sparse spectral domain defined by $\boldsymbol{U}_\ell$, perform GNN computations on a significantly

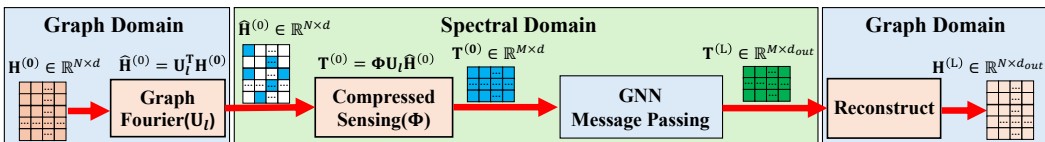

Figure 2: Overall architecture of YOSO. The model first uses a learnable Graph Fourier transform $\boldsymbol{U}_\ell$ to project the input $\boldsymbol{H}^{(0)} \in \mathbb{R}^{N \times d}$ into a sparse representation $\hat{\boldsymbol{H}}^{(0)} \in \mathbb{R}^{N \times d}$. Then applies a fixed sensing matrix $\boldsymbol{\Phi} \in \mathbb{R}^{M \times N}$ ($M \ll N$) to compress the features. YOSO perform the Message Passing directly on the compressed $M \times d$ representation $\boldsymbol{T}^{(0)}$. At output layer, it reconstructs the full-resolution embeddings $\boldsymbol{H}^{(L)} \in \mathbb{R}^{N \times d_{out}}$ to produce task-specific predictions. We should note that, during the training process, Graph Fourier transform $\boldsymbol{U}_l$ is learnable, while the compressed sensing matrix $\boldsymbol{\Phi}$ remains fixed.

compressed $M$-row representation $\boldsymbol{T}^{(l)}$ ($l = [0, 1, ..., L]$) within this domain, and then reconstruct the full $N$-dimensional embeddings $\boldsymbol{H}^{(L)}$ only at the output layer for task-specific predictions and loss calculation. This approach contrasts with the traditional GNNs by primarily operating in a much lower-dimensional space. The key stages are:

**Initialization.** Two sets of parameters are initialized: (1) The learnable orthonormal transformation matrix $\boldsymbol{U}_\ell \in \mathbb{R}^{N \times N}$. This matrix aims to form a Graph Fourier basis where feature matrix become sparse. Thereby $\boldsymbol{U}_\ell$ can be initialized as a random orthogonal matrix (e.g., via QR decomposition of a Gaussian matrix) or using other Stiefel manifold initialization techniques [11], and (2) The GNN model weights $\Theta = \{\boldsymbol{W}^{(l)} \in \mathbb{R}^{d^{(l-1)} \times d^{(l)}}\}_{l=1}^{L}$. These are initialized using standard practices.

**Learnable Spectral Projection (via $\boldsymbol{U}_\ell$).** The input feature matrix $\boldsymbol{H}^{(0)} \in \mathbb{R}^{N \times d}$ is projected into the spectral domain defined by the current learnable basis $\boldsymbol{U}_\ell$:

$$\hat{\boldsymbol{H}}^{(0)} = \boldsymbol{U}_\ell^\top \boldsymbol{H}^{(0)} \tag{3}$$

where $\hat{\boldsymbol{H}}^{(0)} \in \mathbb{R}^{N \times d}$ represents the spectral coefficients of features $\boldsymbol{H}^{(0)}$. To overcome the high cost of using fixed $\boldsymbol{U}$ (as discussed in Obstacle I, Section 3), in YOSO, $\boldsymbol{U}_\ell$ is learned jointly with the GNN parameters $\Theta$, subject to an orthonormality constraint ($\boldsymbol{U}_\ell^\top \boldsymbol{U}_\ell = \boldsymbol{I}$). This learning process (detailed in Section 4.2) guides $\boldsymbol{U}_\ell$ to become a suitable basis that can project node signals (input features and, implicitly, subsequent embeddings) into effectively sparse representations, facilitating efficient compression and high-fidelity reconstruction.

**Compressed Measurement (via $\boldsymbol{\Phi}$).** After obtaining the spectral representation $\hat{\boldsymbol{H}}^{(0)}$, YOSO applies a carefully designed sensing matrix $\boldsymbol{\Phi} \in \mathbb{R}^{M \times N}$ (where $M \ll N$) to generate a compressed version of these spectral coefficients, which forms the initial input $\boldsymbol{T}^{(0)}$ for the GNN layers:

$$\boldsymbol{T}^{(0)} = \boldsymbol{\Phi} \hat{\boldsymbol{H}}^{(0)} \tag{4}$$

$\boldsymbol{T}^{(0)} \in \mathbb{R}^{M \times d}$ is a compressed feature matrix, representing $M$ measurements or intuitively, a sketches of $N$ spectral components. The construction of $\boldsymbol{\Phi}$ is detailed in Section 4.3. The sequence $\boldsymbol{H}^{(0)} \xrightarrow{\boldsymbol{U}_\ell^\top(\cdot)} \hat{\boldsymbol{H}}^{(0)} \xrightarrow{\boldsymbol{\Phi}(\cdot)} \boldsymbol{T}^{(0)}$ is performed at the beginning of each training iteration, as $\boldsymbol{U}_\ell$ updates.

**Forward Propagation in Compressed Domain.** Given compressed representation $\boldsymbol{T}^{(0)} \in \mathbb{R}^{M \times d}$ as input, YOSO performs $L$ layers of message passing entirely within this $M$-dimensional compressed space. For layers $l = 1, \ldots, L$, the compressed embeddings $\boldsymbol{T}^{(l)} \in \mathbb{R}^{M \times d^{(l)}}$ are updated as:

$$\boldsymbol{T}^{(l)} = \sigma\left(\mathcal{A}_{\boldsymbol{\Phi}} \boldsymbol{T}^{(l-1)} \boldsymbol{W}^{(l)}\right) \tag{5}$$

where $\boldsymbol{W}^{(l)}$ is trainable weight matrix for layer $l$, $\sigma(\cdot)$ is a non-linear activation, and $\mathcal{A}_{\boldsymbol{\Phi}} \in \mathbb{R}^{M \times M}$ is a graph propagation operator in the compressed domain. A principled choice for $\mathcal{A}_{\boldsymbol{\Phi}}$ is $\boldsymbol{\Phi} \tilde{\boldsymbol{A}}_{\text{norm}} \boldsymbol{\Phi}^\top$, where $\tilde{\boldsymbol{A}}_{\text{norm}}$ is a normalized adjacency matrix of the original graph (e.g., GCN's $\tilde{\boldsymbol{D}}^{-\frac{1}{2}} \tilde{\boldsymbol{A}} \tilde{\boldsymbol{D}}^{-\frac{1}{2}}$ [36]). This operator allows structural information from the original graph to influence propagation within compressed $M \times M$ space. The weight matrices $\boldsymbol{W}^{(l)}$ thus learn the transformations (analogous to spectral filtering when $\boldsymbol{U}_\ell$ captures frequency components) on these compressed embeddings. After $L$ layers, we obtain the final compressed output $\boldsymbol{T}^{(L)} \in \mathbb{R}^{M \times d_{\text{out}}}$.

**Reconstruction, Loss Computation, and Joint Optimization Objective.** The compressed GNN output $\boldsymbol{T}^{(L)}$ (Eq.(5)), being $M$-dimensional, does not directly correspond to the $N$ original nodes. To bridge this gap and compute a meaningful task-specific loss, YOSO reconstructs an estimate of

the full $N$-dimensional node embeddings $\boldsymbol{H}_c^{(L)*} \in \mathbb{R}^{N \times d_{\text{out}}}$. This reconstruction and the subsequent loss calculation are integrated within a joint optimization objective. Total loss $\mathcal{L}_{\text{total}}$ is defined as:

$$\mathcal{L}_{\text{total}} = \beta \mathcal{L}_{\text{task}}(\boldsymbol{H}_c^{(L)*}, Y_{\text{true}}) + \alpha \mathcal{L}_{\text{recon}}(\boldsymbol{T}^{(L)}, \boldsymbol{\Phi}, \boldsymbol{U}_\ell, \hat{\boldsymbol{H}}_c^{(L)*}, \lambda) \tag{6}$$

where $\boldsymbol{H}_c^{(L)*} = \boldsymbol{U}_\ell \hat{\boldsymbol{H}}_c^{(L)*}$, and term $\hat{\boldsymbol{H}}_c^{(L)*}$ represents the estimated sparse spectral coefficients of the final layer's embeddings. It is not a parameter learned via direct backpropagation in the same way as $\boldsymbol{U}_\ell$ or $\Theta$. Instead, for current training iteration's GNN output $\boldsymbol{T}^{(L)}$ and $\boldsymbol{U}_\ell$, $\hat{\boldsymbol{H}}_c^{(L)*}$ is directly computed by solving the compressed sensing recovery problem (Eq.(1)). The two components of the $\mathcal{L}_{\text{total}}$ are then:

**Task-specific Loss $\mathcal{L}_{\text{task}}$.** This term directly relates to the GNN's predictive performance on the downstream task (e.g., cross-entropy for node classification), computed using the reconstructed $N$-dimensional embeddings $\boldsymbol{H}_c^{(L)*} = \boldsymbol{U}_\ell \hat{\boldsymbol{H}}_c^{(L)*}$ and ground true labels $Y_{\text{true}}$.

**Compressed Sensing Reconstruction Loss $\mathcal{L}_{\text{recon}}$.** This term evaluates how well the GNN output $\boldsymbol{T}^{(L)}$ and the learned basis $\boldsymbol{U}_\ell$ conform to the compressed sensing model using the computed $\hat{\boldsymbol{H}}_c^{(L)*}$. It is defined as:

$$\frac{1}{2} \left\| \boldsymbol{T}^{(L)} - \boldsymbol{\Phi} \boldsymbol{U}_\ell \hat{\boldsymbol{H}}_c^{(L)*} \right\|_F^2 + \lambda \left\| \hat{\boldsymbol{H}}_c^{(L)*} \right\|_{2,1} \tag{7}$$

The first part is the data consistency term, measuring how well the sensed version of the recovered sparse coefficients matches the GNN's compressed output $\boldsymbol{T}^{(L)}$. The second part is the sparsity-promoting regularizer. Both parts of $\mathcal{L}_{\text{recon}}$ explicitly depend on $\hat{\boldsymbol{H}}_c^{(L)*}$, which itself is a function of $\boldsymbol{T}^{(L)}, \boldsymbol{\Phi}, \boldsymbol{U}_\ell$. The hyperparameters $\beta, \lambda, \alpha$ balance these objectives. The overall objective (Eq.(7)) function is then minimized subject to the constraint $\boldsymbol{U}_\ell^\top \boldsymbol{U}_\ell = \boldsymbol{I}$.

## 4.2 Learning the Orthonormal Transformation $\text{U}_l$ and GNN Parameters

As established in Section 4.1, YOSO's training is driven by the joint optimization objective presented in Eq.(7). This section further clarifies how the gradients of this total loss are used to concurrently update the the learnable orthonormal basis $\boldsymbol{U}_\ell$ and GNN parameters $\Theta$.

In each training iteration, after the GNN produces the compressed output $\boldsymbol{T}^{(L)}$, $\hat{\boldsymbol{H}}_c^{(L)*}$ are directly computed by applying FISTA [4] algorithm (Eq.(1)). This $\hat{\boldsymbol{H}}_c^{(L)*}$ is then used to calculate the values for both $\mathcal{L}_{\text{task}}$ (via $\boldsymbol{H}_c^{(L)} = \boldsymbol{U}_\ell \hat{\boldsymbol{H}}_c^{(L)*}$) and $\mathcal{L}_{\text{recon}}$. Consequently, the total loss $\mathcal{L}_{\text{total}}$ becomes a differentiable function of $\Theta$ (through $\boldsymbol{T}^{(L)}$ which is an input to the Eq.(1)) and $\boldsymbol{U}_\ell$ (which is an input to the Eq.(1) and also used in $\mathcal{L}_{\text{task}}$ and $\mathcal{L}_{\text{recon}}$). Automatic differentiation can then compute $\nabla_\Theta \mathcal{L}_{\text{total}}$ and $\nabla_{\boldsymbol{U}_\ell} \mathcal{L}_{\text{total}}$, encouraging the GNN to produce compressed outputs $\boldsymbol{T}^{(L)}$ that lead to low task error and are well-suited for sparse recovery. Theoretical analysis concerning error bounds is in Appendix B.3.

## 4.3 Constructing the Universal Sensing Matrix $\Phi$

A central challenge (Obstacle I, Section 3) in YOSO is designing the fixed sensing matrix $\boldsymbol{\Phi} \in \mathbb{R}^{M \times N}$ to be effective with the learned, evolving basis $\boldsymbol{U}_\ell$ (in Section 4.2). $\boldsymbol{\Phi}$ must robustly capture salient information from the spectral coefficients $\hat{\boldsymbol{H}}^{(0)} = \boldsymbol{U}_\ell^\top \boldsymbol{H}^{(0)}$ and ensure that the combined operation $\boldsymbol{\Phi} \boldsymbol{U}_\ell$ satisfies favorable compressed sensing properties like RIP.

YOSO constructs $\boldsymbol{\Phi}$ by combining a graph-structure-aware component $\boldsymbol{S}_{\text{struct}} \in \mathbb{R}^{M \times N}$, with a randomized component $\boldsymbol{\Sigma}_{\text{rand}} \in \mathbb{R}^{M \times N}$, using element-wise product: $\boldsymbol{\Phi} = \boldsymbol{S}_{\text{struct}} \otimes \boldsymbol{\Sigma}_{\text{rand}}$. The matrix $\boldsymbol{S}_{\text{struct}}$ is determined once during pre-processing based on graph structure and remains fixed. **Construction of the Structural Matrix $\boldsymbol{S}_{\text{struct}}$.** The matrix $\boldsymbol{S}_{\text{struct}}$ aims to guide sensing towards structurally important parts of the graph, which are assumed to correspond to important spectral information. Its construction is based on sampling nodes according to importance scores derived from the eigenvalues of the symmetric normalized Laplacian $\boldsymbol{L}_{\text{sym}} = \boldsymbol{I} - \boldsymbol{D}^{-\frac{1}{2}} \boldsymbol{A} \boldsymbol{D}^{-\frac{1}{2}}$. Let the $N$ eigenvalues of $\boldsymbol{L}_{\text{sym}}$ be $0 \leq \lambda_0 \leq \lambda_1 \leq \cdots \leq \lambda_{N-1}$. A probability $P(i)$ is defined for each spectral mode $i$ (associated with the eigenvalue $\lambda_i$) as $P(i) = \frac{w(\lambda_i)}{\sum_{j=0}^{N-1} w(\lambda_j)}$ where $w(\lambda_i)$ is a weighting function reflecting the importance of the $i$-th spectral mode. To construct the $M$ rows of $\boldsymbol{S}_{\text{struct}} \in \{0,1\}^{M \times N}$, for each row $k = 1, \ldots, M$: (1) Sample spectral index $i_k \in \{0, \ldots, N-1\}$

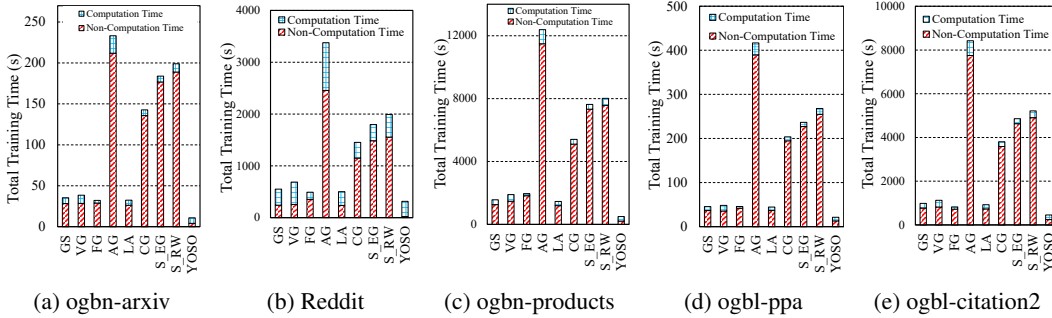

Figure 3: The total training time comparison (with breakdown). The evaluation covers two learning tasks across five datasets [30, 28]: (a) to (c) represent the results for the node classification task on ogbn-arxiv, Reddit, and ogbn-products, respectively; while (d)-(e) correspond to the link prediction task on ogbl-ppa and ogbl-citation2. The schemes are GS (GraphSage [28]), VG (VR-GCN [12]), FG (FastGCN [13]), AG (AS-GCN [31]), LA (LADIES [72]), CG (Cluster-GCN [15]), two versions of GraphSAINT [66] (S_EG and S_RW), and our proposed YOSO.

according to the probability distribution $P(i)$, and (2) The $k$-th row $(\boldsymbol{S}_{\text{struct}})_{k,:}$ is set to be $i_k$-th row of $N \times N$ identity matrix $\boldsymbol{I}_N$. That is, $(\boldsymbol{S}_{\text{struct}})_{k,j} = 1$ if $j = i_k$, and 0 otherwise. This construction means each row of $\boldsymbol{S}_{\text{struct}}$ selects exactly one spectral coefficient based on the eigenvalue-weighted sampling. Thus, $\boldsymbol{S}_{\text{struct}}$ is a binary matrix where each row has exactly one non-zero entry. This inherently ensures that $\boldsymbol{S}_{\text{struct}}$ has no all-zero rows (provided $M > 0$). This structural property is important for the full rank characteristic of $\boldsymbol{\Phi}$, as detailed in Appendix B.1.

**Construction of the Random Matrix $\boldsymbol{\Sigma}_{\text{rand}}$.** Randomness is introduced via $\boldsymbol{\Sigma}_{\text{rand}} \in \mathbb{R}^{M \times N}$ to help satisfy RIP-like conditions [3]. The construction of $\boldsymbol{\Sigma}_{\text{rand}}$ is coordinated with $\boldsymbol{S}_{\text{struct}}$. For each column $j = 1, \ldots, N$ of $\boldsymbol{S}_{\text{struct}}$, let $g(j) = \sum_{k=1}^{M} (\boldsymbol{S}_{\text{struct}})_{k,j}$ be the number of non-zero elements in that column (i.e., how many sampled "sensor neighborhoods" node $j$ is part of). The elements of $\boldsymbol{\Sigma}_{\text{rand}}$ are defined as: for each entry $(\boldsymbol{\Sigma}_{\text{rand}})_{k,j}$,

$$(\boldsymbol{\Sigma}_{\text{rand}})_{k,j} \sim \begin{cases} \mathcal{N}(0, \frac{1}{g(j)}) & \text{if } (\boldsymbol{S}_{\text{struct}})_{k,j} = 1 \text{ and } g(j) > 0 \\ 0 & \text{if } (\boldsymbol{S}_{\text{struct}})_{k,j} = 0 \end{cases}$$

If $g(j) = 0$ for some $j$ (meaning node $j$ is not in any sampled neighborhood), then $(\boldsymbol{\Sigma}_{\text{rand}})_{k,j} = 0$ for all $k$. This implies the $j$-th column of $\boldsymbol{\Phi}$ will be all zeros. The specific design for $\boldsymbol{\Phi} = \boldsymbol{S}_{\text{struct}} \otimes \boldsymbol{\Sigma}_{\text{rand}}$ combines the graph-aware structural selection (via node importance and neighborhoods encoded in $\boldsymbol{S}_{\text{struct}}$) with scaled randomness (in $\boldsymbol{\Sigma}_{\text{rand}}$). It aims to create a fixed sensing matrix that is effective for capturing information from spectrally sparse signals generated by the evolving $\boldsymbol{U}_\ell$. The claim that this construction satisfies Restricted Isometry Property is detailed in Appendix B.2.

## 5 Experiments

We evaluate YOSO and other baselines with across two most widely used tasks: node classification and link prediction. For the detailed description of experimental setting, such as the datasets (Please refer to Table 5 for the detailed descriptions of the datasets), baselines and hyperparameter, are provided in Appendix A.

### 5.1 Overall Comparison

We evaluate baselines and YOSO with two core metrics: model accuracy and total training time.

**Node Classification Task.** YOSO achieves the shortest training time with an average of 74% reduction among all baselines as shown in Fig. 3. For most of the cases, YOSO can achieve more than 60% training time reduction. For two cases with less than 40% reduction compared to FG and LA on Reddit. The training time is reduced from around 490s with FG and around 501s with LA to around 341s with YOSO (with 36% and 37% time reduction, respectively). The main reason is that YOSO can significantly reduce the Non-computation time while introducing a little reconstruction overhead. On average, YOSO reduces non-computation time by approximately 95.7% compared to all other baselines. For model accuracy shown in Table 1, YOSO consistently matches or closely approaches the top performers. For example, YOSO obtains an accuracy of 0.71 on ogbn-arxiv, just 0.01 below GraphSage.

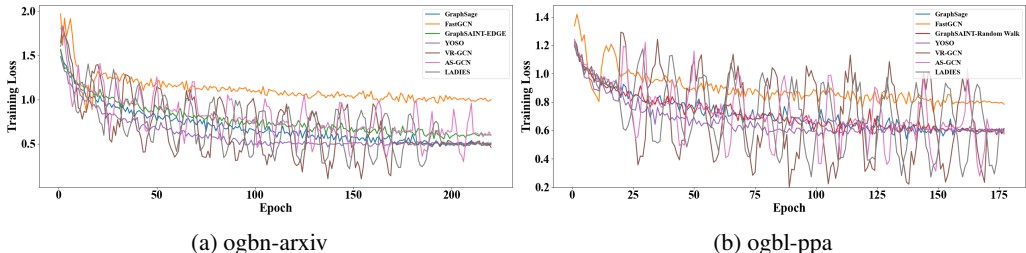

| (a) ogbn-arxiv | (b) ogbl-ppa |

Figure 4: Training loss and epoch curves for YOSO and baselines on two benchmark datasets.

**Link Prediction Task.** For total training time, similar to the node classification task, YOSO achieves the best training time with around $72\%$ average training time decrease across all datasets for the link prediction. This improvement is consistent with the node classification task, where YOSO achieves considerable reductions in non-computation time while introducing minimal reconstruction overhead. As shown in Fig. 3(d)-(e), YOSO achieves an average non-computation time reduction of about $81\%$ across all datasets. As for model accuracy, in Table 1, YOSO maintained results with only a very small gap: $0.003$ on ogbn-arxiv and ogbl-citation2, compared to the best results.

Table 1: Model accuracy results for different methods on node classification and link prediction tasks. Accuracy metrics on each dataset are in Table 5.

| Methods | Datasets | | | | |
|---|---|---|---|---|---|
| | Node Classification | | | Link Prediction | |
| | arxiv | Reddit | products | ppa | citation2 |
| GraphSage | **0.720** | 0.949 | 0.772 | 0.170 | 0.805 |
| VR-GCN | 0.697 | 0.962 | 0.699 | 0.170 | 0.796 |
| FastGCN | 0.438 | 0.927 | 0.404 | 0.108 | 0.655 |
| AS-GCN | 0.687 | 0.964 | 0.510 | 0.124 | 0.659 |
| LADIES | 0.649 | 0.927 | 0.501 | 0.113 | 0.669 |
| Cluster-GCN | 0.653 | 0.966 | 0.769 | 0.205 | 0.790 |
| GraphSAINT-EG | 0.702 | **0.967** | **0.792** | 0.214 | 0.804 |
| GraphSAINT-RW | 0.701 | **0.967** | 0.783 | **0.226** | **0.805** |
| YOSO | **0.720** | **0.967** | 0.787 | 0.223 | 0.802 |

In summary, for both tasks of node classification and link prediction, by combining high accuracy with substantial reductions in sampling and total training time, YOSO demonstrates its efficiency in GNN training and significantly improves both sampling and total training times across all datasets while maintaining competitive accuracy, highlighting its effectiveness compared to the baselines on the node classification task.

## 5.2 Convergence Comparison

We investigate YOSO convergence performance compared to other baselines on ogbn-arxiv (node classification) and ogbl-ppa (link prediction). The training loss-epoch curves are shown in Fig. 4. In two different learning tasks, YOSO consistently outperformed the baselines in terms of convergence speed and stability. For ogbn-arxiv, YOSO reached a lower training loss more rapidly than Graph-SAGE, GraphSAINT-EDGE, and FastGCN, with significantly fewer oscillations, indicating a more stable and efficient training process. Similarly, for ogbl-ppa, YOSO demonstrated faster convergence and maintained a smoother training loss curve compared to other baselines. These results suggest that YOSO not only accelerates the convergence process but also ensures a more stable training path compared to existing sampling methods, highlighting its effectiveness in GNN training.

## 5.3 Ablation Study

We explore three main aspects of YOSO design: (i) How YOSO's total training time and model accuracy vary with different size of $M$, (ii) Reconstruction effectiveness by comparing the $\boldsymbol{H}^{(L)}$ and $\hat{\boldsymbol{H}}^{(L)}$, and (iii) The parameterization strategy of $\boldsymbol{\Phi}$ (layer-wise **vs.** universal) (see Section 4.3).

**Varying $M$ value.** We examine how total training time (including breakdown) and model accuracy vary with $M$ values, specifically $M = \{64, 128, 256, 1024, 2048\}$, as shown in Fig. 5. The results indicate that YOSO's sampling time remains stable across different $M$, ranging from 107.94 to 111.53 seconds on ogbn-products and 143.56 to 149.65 seconds on ogbl-citation2, showing minimal impact from $M$. In contrast, as $M$ decreases, computation time increases, reflecting more iterations needed for convergence (e.g., rising from 275.98s at $M = 2048$ to 301.94s at $M = 64$ on ogbn-products, with a similar trend on ogbl-citation2). Model accuracy improves with larger $M$, eventually stabilizing; it rises from 0.597 to 0.7873 on ogbn-products and from 0.312 to 0.8025 on ogbl-citation2. These findings highlight YOSO's efficient sampling and improved accuracy and convergence with larger $M$.

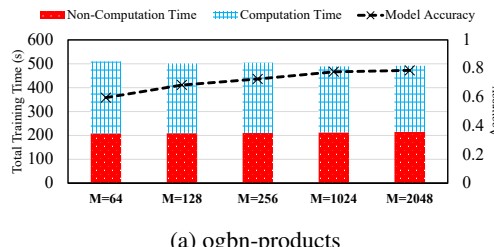
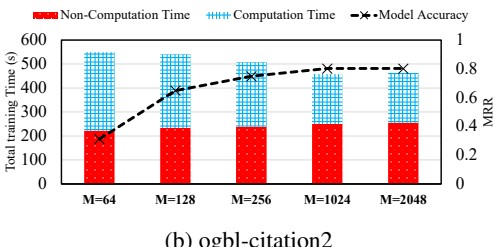

(a) ogbn-products · (b) ogbl-citation2

Figure 5: Total training time (including its breakdown) and model accuracy for YOSO with different sampling sizes: (a) for the node classification learning task on the ogbn-products dataset, and (b) for the link prediction learning task on the ogbl-citation2 dataset.

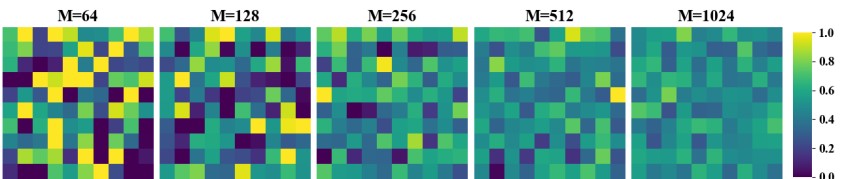

Figure 6: Reconstruction effectiveness visualized via heatmap. Using the ogbn-products dataset, 10 nodes are randomly selected from the training set, and for each node, 10 embedding dimensions are randomly picked. The heatmap shows the absolute differences between original and reconstructed embeddings for these elements. $M$ is the size of the sampling set.

Table 2: Total training time and model accuracy under different $\Phi$ configurations.

| Dataset | ogbn-arxiv | | ogbl-ppa | |
|---|---|---|---|---|
| Type of $\Phi$ | Layer-wise | Universal | Layer-wise | Universal |
| Total Training Time (s) | 59.22 | 10.93 | 145.50 | 21.46 |
| Model Accuracy | 0.730 | 0.727 | 0.2254 | 0.2235 |

**Reconstruction effectiveness:** Fig. 6 shows the reconstruction effectiveness for different sampling sizes $M$. Each $10 \times 10$ block represents the absolute difference between reconstructed embeddings from our two-layer GNN sampling and those computed with all neighbors (without sampling). As $M$ increases, reconstruction accuracy improves, enhancing overall model accuracy. However, beyond a certain point, such as $M = 512$ in Fig. 6, further increases in $M$ offer diminishing returns in both reconstruction quality and model accuracy. This suggests there is an optimal $M$ that balances reconstruction quality and computational efficiency.

**Layer-wise vs. Universal.** We compare two parameterization strategies: a layer-wise variant, where each GNN layer maintains its own transformation matrix, and a universal variant that shares a single $\Phi$ across layers. As shown in Table 2, the layer-wise $\Phi$ notably increases computational overhead: by approximately $5\times$ on ogbn-arxiv and $7\times$ on ogbl-ppa, while providing only marginal accuracy improvements (within 0.001). This indicates that a universal $\Phi$ achieves nearly equivalent accuracy at substantially lower cost, offering a favorable efficiency–accuracy trade-off.

## 5.4 Comparisons to Graph Condensation/Distillation and Linearization

To further contextualize YOSO's efficiency, we compare it with two other families of graph-level optimization schemes that also aim to reduce the training cost: (i) Graph Condensation/Distillation methods, which synthesize smaller representative graphs for training, and (ii) Linearization methods, which approximate message passing through pre-computed feature propagation.

**YOSO vs. Graph Condensation/Distillation.** Both Graph Condensation and Graph Distillation introduce the substantial preprocessing overhead and risk discarding informative structural signals, which can degrade model performance. To quantify this, we compare YOSO against two most widely used condensation schemes: GCond [34] and GC-SNTK [60], on the ogbn-arxiv dataset with a 0.25% reduction ratio. As shown in the Table 3, YOSO achieves higher accuracy (0.7169) while requiring

Table 3: Comparison of preprocessing time and model accuracy on the ogbn-arxiv dataset

| Dataset | ogbn-arxiv | | |
|---|---|---|---|
| Schemes | GCond | GC-SNTK | YOSO |
| Preprocessing Time (s) | 20615.6 | 11066.89 | 1643.32 |
| Model Accuracy | 0.6172 | 0.6219 | 0.7169 |

Table 4: Comparison between YOSO with linearization schemes.

| Dataset | ogbn-products | | | | | ogbn-arxiv | | |
|---|---|---|---|---|---|---|---|---|
| Schemes | SIGN-2 | SIGN-4 | SIGN-6 | SIGN-8 | YOSO | iSVD | iSVD-best | YOSO |
| Total Training Time (s) | 421.79 | 584.07 | 831.94 | 1052.96 | 499.02 | 9.94 | 982.12 | 10.74 |
| Model Accuracy | 0.761 | 0.778 | 0.776 | 0.783 | 0.788 | 0.685 | 0.746 | 0.720 |

far less preprocessing time ($12\times$ faster than GCond and $6\times$ faster than GC-SNTK), demonstrating superior efficiency and information retention.

**YOSO *vs.* Linearization.** We also compare YOSO with multiple linearized GNN variants, including SIGN [22] and iSVD [1], on ogbn-products and ogbn-arxiv. As summarized in the Table 4, SIGN-2 achieves slightly lower training time but at the cost of a 2.7% accuracy drop. The low-accuracy iSVD variant reduces runtime by 8% yet loses 4% accuracy, whereas the high-accuracy version (iSVD-best) increases runtime by $91\times$ for only a 0.014 gain. YOSO consistently delivers a better balance between accuracy and total training time.

# 6 Related Work

Numerous approaches have been proposed to enhance GNN training efficiency. Sampling-based methods, including node-wise, layer-wise, and subgraph-based techniques [28, 13, 15], aim to reduce computational load by training on smaller portions of the graph. While effective in certain scenarios, node-wise sampling can suffer from exponentially growing receptive fields, layer-wise methods may introduce bias or complex variance reduction schemes, and subgraph-based approaches often incur significant pre-processing overhead for graph partitioning or dynamic subgraph generation, sometimes at the cost of accuracy (e.g., layer-wise schemes) or increased complexity (e.g., subgraph-based). Other strategies such as graph condensation and distillation [34, 56] focus on creating smaller graph proxies or simpler models, which can be effective but may lose fine-grained information or require careful tuning of the condensation/distillation process itself. Meanwhile, historical embedding [12] and linearization techniques [22] seek to optimize specific aspects of GNN computation or simplify model architecture, often by removing non-linearities which can limit expressive power. YOSO distinguishes itself by uniquely combining a once-per-training learnable orthonormal spectral transformation with Compressed Sensing principles for a one-shot projection and reconstruction. This approach primarily targets the reduction of non-computational overheads (like extensive data handling for sampling and repeated transformations) without resorting to complex graph sampling strategies or repeated costly operations during the GNN's forward and backward passes. A more detailed discussion of related work is provided in Appendix C.

# 7 Conclusion

In this paper, we introduce YOSO (You Only Spectralize Once), a novel training scheme aimed at significantly enhancing the efficiency of GNN training without sacrificing prediction accuracy. By leveraging a Graph Fourier and compressed sensing-based reconstruction framework, YOSO performs a spectral transformation only once at the input layer, followed by an error-bounded reconstruction at the output layer during each training iteration. Our experimental results demonstrate that YOSO can achieve an average 74% reduction of existing state-of-the-art schemes while preserve model accuracy comparable to top-performing baselines.

## Acknowledgment

This work was partially supported by NSF 2204656, 2343863, 2413520 and 2440611. Any opinions, conclusions, or the recommendations expressed in this material are those of the authors and do not necessarily reflect the views of the NSF.

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

# A   Experimental Setting

## A.1   Hardware and Software Configuration

We evaluate all baselines and YOSO on Ubuntu 18.04.6 LTS, equipped with an NVIDIA GTX 1060Ti (6GB memory), using CUDA version 11.8 and PyTorch version 2.0.0. The system features a AMD Ryzen 5 5900 CPU with 128 GB DDR4 RAM, and the Python version used is 3.9.0.

Table 5: Statistics and metrics of the dataset.

| Dataset | | #node | #edge | #dim. | Metric |
|---|---|---|---|---|---|
| Node Property Prediction | ogbn-arxiv | 169,343 | 1,166,243 | 128 | Accuracy |
| | Reddit | 232,965 | 11,606,919 | 602 | Micro-F1 |
| | ogbn-products | 2,499,029 | 61,859,140 | 100 | Accuracy |
| Link Property Prediction | ogbl-ppa | 576,289 | 30,326,273 | 128 | Hits@100 |
| | ogbl-citation2 | 2,927,963 | 30,561,187 | 128 | MRR |

## A.2   Datasets

For node classification, we use Reddit [28], ogbn-arxiv and ogbn-products [30] and for link prediction, we use ogbl-ppa, and ogbl-citation2 [30]. The basic summary information of the datasets we use is provided in Table 5, and detailed descriptions are as follows:

**ogbn-arxiv**: This dataset is a directed citation network of Computer Science (CS) arXiv papers from the Microsoft Academic Graph (MAG) [59]. Each node represents a paper, with directed edges indicating citations. The task is to classify unlabeled papers into primary categories using labeled papers and node features, which are derived by averaging word2vec embeddings [43] of paper titles and abstracts.

**Reddit**: Originally from GraphSage [28], this Reddit dataset is a post-to-post graph where each node represents a post, and edges indicate shared user comments. The task is to classify posts into communities using GloVe word vectors [47] from post titles and comments, along with features such as post scores and comment counts.

**ogbn-products**: This undirected, unweighted graph represents an Amazon product co-purchasing network, where nodes are products and edges indicate frequent co-purchases. Node features are derived from bag-of-words features of product descriptions, reduced to 100 dimensions via Principal Component Analysis [18].

**ogbl-ppa**: This undirected, unweighted graph has nodes representing proteins from 58 species, with edges indicating biologically meaningful associations. Each node features a 58-dimensional one-hot vector for the protein's species. The task is to predict new association edges, evaluated by ranking positive test edges over negative ones.

**ogbl-citation2**: This dataset is a directed graph representing a citation network among a subset of papers from Microsoft Academic Graph (MAG), similar to ogbn-arxiv. For each source paper, two references are randomly removed, and the task is to rank these missing references above 1,000 randomly selected negative references, which are sampled from all papers not cited by the source paper.

**Data splitting**: We adopt strategies consistent with previous works [28, 30]. Specifically, for the Reddit dataset, we follow the data splitting used in GraphSage [28], and for the OGB series (ogbn and ogbl), we maintain the splitting described in [30].

## A.3   Baselines and Implementation

The baselines used cover all major methods designed to address the trade-off between efficiency and accuracy, including: GraphSage [28] (GS), VR-GCN [12] (VG), FastGCN [13] (FG), AS-GCN [31] (AG), LADIES [72] (noted by LA), Cluster-GCN [15] (CG), and GraphSAINT [66] (two versions noted as S_EG and S_RW). Detailed information on the source code for these baselines, the YOSO implementation, and other related materials can be found in Appendix A.3.

Table 6: Baselines and their public available source code link

| Scheme | Available Link |
|---|---|
| GraphSage | https://github.com/williamleif/graphsage-simple |
| VR-GCN | https://github.com/THUDM/cogdl/tree/master/examples/VRGCN |
| FastGCN | https://github.com/gmancino/fastgcn-pytorch |
| AS-GCN | https://github.com/Gkunnan97/FastGCN_pytorch |
| LADIES | https://github.com/acbull/LADIES |
| Cluster-GCN | https://github.com/benedekrozemberczki/ClusterGCN |
| GraphSAINT | https://github.com/GraphSAINT/GraphSAINT |

Table 7: Node classification hyperparamter setting for baselines and YOSO on different datasets.

| Scheme | ogbn-arxiv | Reddit | ogbn-products |
|---|---|---|---|
| GraphSage | 25&10 / Adam / 0.7 | 25&10 / Adam / 0.01 | 50&20 / Adam / 0.01 |
| VR-GCN | 8 / Adam / 0.01 | 16 / Adam / 0.01 | 32 / Adam / 0.01 |
| FastGCN | 64 / Adam / 0.01 | 128 / Adam / 0.001 | 256 / Adam / 0.001 |
| AS-GCN | 128 / Adam / 0.001 | 512 / Adam / 0.01 | 1000 / Adam / 0.01 |
| LADIES | 64 / Adam / 0.001 | 128 / Adam / 0.001 | 256 / Adam / 0.001 |
| Cluster-GCN | - / Adam / 0.01 | - / Adam / 0.005 | - / Adam / 0.005 |
| GraphSAINT-EG | 300 / Adam / 0.01 | 600 / Adam / 0.01 | 4000 / Adam / 0.01 |
| GraphSAINT-RW | 4000 / Adam / 0.01 | 8000 / Adam / 0.01 | 10000 / Adam / 0.01 |
| YOSO | 128 / Adam / 0.01 | 256 / Adam / 0.01 | 512 / Adam / 0.01 |

Table 6 presents the baselines used in this paper along with their publicly available source code links. Since some baselines were not originally implemented in PyTorch, we standardized the framework for fair comparison. If a PyTorch version involved the original authors, we selected that source code (e.g., FastGCN [13]). Otherwise, we chose the most popular implementation based on the number of stars. Notably, the repository linked for AS-GCN [31] in the table includes implementations of both FastGCN and AS-GCN, but we only used the AS-GCN version, while the FastGCN implementation was taken from the source listed in the table.

**YOSO's Implementation**: The base code of YOSO[1] is built on GCN [36], with the link available at https://github.com/tkipf/pygcn. The sampling stage in YOSO occurs on the CPU and main memory since it involves calculations related to the entire feature matrix and the regularized Laplacian matrix. After sampling, the relevant data is migrated to GPU memory for computation. Throughout the training process, multiple data exchanges occur between main memory and GPU memory, such as in link prediction tasks where node embeddings need to be updated.

**Modification:** All baselines support updating node embeddings and performing node classification tasks. For node classification, if a baseline did not originally use the cross-entropy loss function, we adjusted it to adopt this loss function. For the link prediction task, the following loss function is applied:

$$\mathcal{L} = \frac{1}{N^+} \sum_{(i,j) \in E^+} \left( 1 - \frac{\boldsymbol{h}_i^{(L)} \cdot \boldsymbol{h}_j^{(L)}}{\|\boldsymbol{h}_i^{(L)}\|\|\boldsymbol{h}_j^{(L)}\|} \right) + \frac{1}{N^-} \sum_{(i,j) \in E^-} \max \left( 0, \gamma - \left( 1 - \frac{\boldsymbol{h}_i^{(L)} \cdot \boldsymbol{h}_j^{(L)}}{\|\boldsymbol{h}_i^{(L)}\|\|\boldsymbol{h}_j^{(L)}\|} \right) \right)$$

where $N^+$ and $N^-$ represent the number of positive and negative samples, respectively, and $E^+$ and $E^-$ denote the sets of positive and negative edges. The parameter $\gamma$ is a hyperparameter, set to 0.5 in this study. As the ogbl-ppa and ogbl-citation2 datasets provide corresponding negative edges by default, we used these pre-defined negative edges for our calculations.

## A.4 Hyper-parameter Setting

All experiments are conducted with two layers. The hyperparameter settings for both YOSO and the baselines are provided in Table 7 and Table 8 for node classification and link prediction datasets, respectively. All experiments were conducted using a two-layer GCN with official configurations.

---

[1]https://anonymous.4open.science/r/YOSO-B49B

Table 8: Link prediction hyperparamter setting for baselines and YOSO on different datasets.

| Scheme | ogbl-ppa | ogbl-citation2 |
|---|---|---|
| GraphSage | 25&10 / Adam / 0.7 | 50&20 / Adam / 0.01 |
| VR-GCN | 8 / Adam / 0.01 | 32 / Adam / 0.01 |
| FastGCN | 64 / Adam / 0.01 | 256 / Adam / 0.001 |
| AS-GCN | 128 / Adam / 0.001 | 1000 / Adam / 0.01 |
| LADIES | 64 / Adam / 0.001 | 256 / Adam / 0.001 |
| Cluster-GCN | - / Adam / 0.01 | - / Adam / 0.005 |
| GraphSAINT-EG | 300 / Adam / 0.01 | 4000 / Adam / 0.01 |
| GraphSAINT-RW | 4000 / Adam / 0.01 | 10000 / Adam / 0.01 |
| YOSO | 128 / Adam / 0.01 | 512 / Adam / 0.01 |

When certain parameters were not clearly specified in some papers, we fine-tuned them for optimal accuracy. The recorded hyperparameters include the sampling size (per node/layer/subgraph), the optimizer, and the learning rate. For YOSO, the sampling size is denoted as $M$; for example, on the ogbl-ppa dataset (Table 8), $M = 128$.

## B  Formal Proof

### B.1  Full Rank of $\Phi$

**Theorem 1 (Full Rank of $\Phi$):** Let $\mathbf{S}_{struct} \in \{0,1\}^{M \times N}$ be the structural matrix where each of its $M$ rows, $k$ is constructed by sampling a spectral index $i_k \in 0, ..., (N-1)$ according to the probability distribution $P(i) = w(\lambda_i)/\sum_{j=0}^{N-1} w(\lambda_j)$ (with $w(\lambda_i) = \lambda_i$ as proposed in Section 4.3), and setting $(\mathbf{S}_{struct})_{k,j} = 1$ if $j = i_k$ and 0 otherwise. Assume the $M$ sampled spectral indices $i_k$ are distinct (ie., sampling without replacement, requiring $M \leq N$). Let $\mathbf{\Sigma}_{rand} \in \mathbb{R}^{M \times N}$ be a random matrix where entries $(\mathbf{\Sigma}_{rand})_{k,j}$ are defined as $(\mathbf{\Sigma}_{rand})_{k,j} \sim \mathcal{N}(0, 1/g(j))$ if $(\mathbf{S}_{struct})_{k,j} = 1$ and $g(j) > 0$, and $(\mathbf{\Sigma}_{rand})_{k,j} = 0$ if $(\mathbf{S}_{struct})_{k,j} = 0$, where $g(j) = \sum_{k=1}^{M} (\mathbf{S}_{struct})_{k,j}$. Define the sensing matrix $\mathbf{\Phi} = \mathbf{S}_{struct} \otimes \mathbf{\Sigma}_{rand}$, where $\otimes$ denotes element-wise multiplication. Then, with probability 1, the matrix $\mathbf{\Phi}$ has full row rank $M$.

**Proof:** The structure of $\mathbf{\Phi}$ is such that for each row $k \in 1, ..., M$, only one entry is non-zero. Specifically, if the spectral index $i_k$ was chosen for row $k$ (so $(\mathbf{S}_{struct})_{k,i_k} = 1$ and $(\mathbf{S}_{struct})_{k,j} = 0$ for $j \neq i_k$), then $\mathbf{\Phi}_{k,i_k} = (\mathbf{S}_{struct})_{k,i_k} \cdot (\mathbf{\Sigma}_{rand})_{k,i_k} = 1$, and all other entries $(\mathbf{\Sigma}_{rand})_{k,j}$ for $j \neq i_k$ are 0. Since we assume the $M$ sampled spectral indices $i_1, i_2, ..., i_M$ are distinct, it follow that for each selected index $i_k$, $g(i_k) = \sum_{r=1}^{M} (\mathbf{S}_{struct})_{r,i_k} = 1$ (as only row $k$ has '1' in column $i_k$). Thus, for the non-zero entry in row $k$, $i_k = (\mathbf{\Sigma}_{rand})_{k,i_k} \sim \mathcal{N}(0, 1/1) = \mathcal{N}(0, 1)$. To show that $\mathbf{\Phi}$ has full row rank $M$, we need to demonstrate that its $M$ rows are linearly independent. Consider a linear combination of the rows of $\mathbf{\Phi}$ that equals the zero row vector $\mathbf{0}^{\top} \in \mathbb{R}^{1 \times N}$:

$$\sum_{k=1}^{M} c_k (\mathbf{\Phi})_{k,:} = \mathbf{0}^{\top} \tag{8}$$

where $c_k$ are scalars and $(\mathbf{\Phi})_{k,:}$ is the $k$-th row of $\mathbf{\Phi}$. This vector equation implies that for each column $j \in 0, ..., N-1$:

$$\sum_{k=1}^{M} c_k (\mathbf{\Phi})_{k,j} = 0 \tag{9}$$

Consider one of the $M$ distinct spectral indices that were sampled, say $i_p$ (where $p \in 1, ..., M$) is the row for which this index was chosen). For the column $j = i_p$, the sum becomes:

$$c_p (\mathbf{\Phi})_k \neq p c_k (\mathbf{\Phi})_{k,i_p} = 0 \tag{10}$$

since $(\mathbf{\Phi})_{k,i_p} = 0$ for all $k \neq p$ (because the sampled indices $i_k$ are distinct, so only row $p$ has its non-zero entry in column $i_p$), the equation simplifies to:

$$c_p (\mathbf{\Phi})_{p,i_p} = 0 \tag{11}$$

as $(\mathbf{\Phi})_{p,i_p} \sim \mathcal{N}(0,1)$, it is non-zero with probability 1. Therefore, for the equation $c_p(\mathbf{\Phi})_{p,i_p} = 0$ to hold, we must have $c_p = 0$. Since this argument applies to each of the $M$ distinct sampled indices $i_1, ..., i_M$, it follows that all coefficients $c_1, c_2, ..., c_M$ must be zero. This demonstrates that the rows of $\mathbf{\Phi}$ are linearly independent with probability 1. Thus, $\text{rank}(\mathbf{\Phi}) = M$.

## B.2 Sensing Matrix $\mathbf{\Phi}$, Learnable $\mathbf{U}_\ell$, and RIP Satisfaction

**Theorem 2 (RIP Satisfaction for $\mathbf{\Phi U}_\ell$):** Let $\mathbf{U}_\ell \in \mathbb{R}^{N \times N}$ be an orthonormal matrix. Let $\mathbf{\Phi} \in \mathbb{R}^{M \times N}$ be constructed as in Theorem 1 (effectively selecting $M$ distinct rows of $\mathbf{U}_\ell$ and multiplying them by independent $\mathcal{N}(0,1)$ scalars). If $\mathbf{U}_\ell$ is sufficiently incoherent with the standard basis (in which $\hat{\mathbf{H}}$ is $k$-row-sparse), then for any $0 < \delta_k < 1$, there exists a constant $C_{RIP} > 0$ such that if $M \geq C_{RIP} \cdot k \log(N/k)$, the matrix $\mathbf{\Psi} = \mathbf{\Phi U}_\ell$ satisfies the $k$-row-RIP for matrices $\hat{\mathbf{H}} \in \mathbb{R}^{N \times d}$ with constant $\delta_k$ with high probability:

$$(1 - \delta_k)\|\hat{\mathbf{H}}\|_F^2 \leq \|\mathbf{\Phi U}_\ell \hat{\mathbf{H}}\|_F^2 \leq (1 + \delta_k)\|\hat{\mathbf{H}}\|_F^2$$

.

**Proof.** Let $\mathbf{A} = \mathbf{\Phi U}_\ell$. The $k$-th row of $\mathbf{A}$ is $(\mathbf{\Sigma})_{k,i_k} \cdot ((\mathbf{U}_\ell)_{i_k,:})^\top$, where $((\mathbf{U}_\ell)_{i_k,:})^\top$ is the $i_k$-th row of $\mathbf{U}_\ell$. The core idea is to show that for any $k$-row-sparse matrix $\hat{\mathbf{H}}$, $\|\mathbf{A}\hat{\mathbf{H}}\|_F^2$ concentrates around $\|\hat{\mathbf{H}}\|_F^2$.

**Step 1: Expectation of the Squared Norm.** Let $\mathbf{X}' = \mathbf{U}_\ell \hat{\mathbf{H}}$. The $k$-th row of $\mathbf{\Phi X}'$ is $(\mathbf{\Phi X}')_{k,:} = (\mathbf{\Sigma})_{k,i_k}(\mathbf{X}')_{i_k,:}$.

$$\mathbb{E}\left[\|\mathbf{\Phi U}_\ell \hat{\mathbf{H}}\|_F^2\right] = \mathbb{E}\left[\sum_{p=1}^{M} \|((\mathbf{\Phi U}_\ell \hat{\mathbf{H}})_{p,:})\|_2^2\right]$$

$$= \sum_{p=1}^{M} \mathbb{E}\left[((\mathbf{\Sigma})_{p,i_p})^2\right] \|((\mathbf{U}_\ell)_{i_p,:})^\top \hat{\mathbf{H}}\|_F^2$$

Since $(\mathbf{\Sigma})_{p,i_p} \sim \mathcal{N}(0,1)$, $\mathbb{E}[((\mathbf{\Sigma})_{p,i_p})^2] = 1$.

$$\mathbb{E}\left[\|\mathbf{\Phi U}_\ell \hat{\mathbf{H}}\|_F^2\right] = \sum_{p=1}^{M} \|((\mathbf{U}_\ell)_{i_p,:})^\top \hat{\mathbf{H}}\|_F^2$$

This is the sum of energies of $\hat{\mathbf{H}}$ projected onto the $M$ selected rows of $\mathbf{U}_\ell$. For this sum to be equal to $\|\hat{\mathbf{H}}\|_F^2 = \|\mathbf{U}_\ell \hat{\mathbf{H}}\|_F^2$ (as $\mathbf{U}_\ell$ is orthonormal), the $M$ selected rows must effectively form a basis for the $k$-row-sparse $\hat{\mathbf{H}}$. This is where incoherence and random selection are crucial. If the selection of $i_p$ were uniformly random and $\mathbf{U}_\ell$ were incoherent with the sparsity basis of $\hat{\mathbf{H}}$, this sum would be proportional to $(M/N)\|\hat{\mathbf{H}}\|_F^2$. The matrix $\mathbf{\Phi}$ needs to be appropriately scaled (e.g., by $1/\sqrt{M}$ or $N/M$) if this expectation is to be exactly $\|\hat{\mathbf{H}}\|_F^2$. The current construction with $\mathcal{N}(0,1)$ weights results in an expectation that is the sum of energies in $M$ selected components. Standard RIP proofs often assume a sensing matrix $\mathbf{A}$ such that $\mathbb{E}[\|\mathbf{A}x\|^2] = \|x\|^2$. For our $\mathbf{\Phi U}_\ell$, we'd typically need a scaling factor of $\sqrt{N/M}$ or similar for the random entries if we were constructing a dense random projection, or ensure the sum of energies of selected components approximates the total energy.

**Step 2: Concentration for a Fixed $k$-row-sparse $\hat{\mathbf{H}}$.** For a fixed $k$-row-sparse $\hat{\mathbf{H}}$, one uses matrix concentration inequalities (e.g., Matrix Bernstein, Hanson-Wright for quadratic forms of random variables, or specific results for random sampling from orthonormal systems) to show that $\|\mathbf{\Phi U}_\ell \hat{\mathbf{H}}\|_F^2$ is tightly concentrated around its expectation.

$$P\left(\left|\|\mathbf{\Phi U}_\ell \hat{\mathbf{H}}\|_F^2 - \mathbb{E}[\|\mathbf{\Phi U}_\ell \hat{\mathbf{H}}\|_F^2]\right| \geq \delta_k \mathbb{E}[\|\mathbf{\Phi U}_\ell \hat{\mathbf{H}}\|_F^2]\right) \leq 2\exp(-c_1 M \delta_k^2 / k)$$

(The exact form of the exponent depends on the specific inequality and properties of $\mathbf{U}_\ell$ and the sampling $P(i)$.) Assuming $\mathbb{E}[\|\mathbf{\Phi U}_\ell \hat{\mathbf{H}}\|_F^2] \approx \|\hat{\mathbf{H}}\|_F^2$ (which requires careful normalization or argument about the selection process effectively capturing the energy of $k$-sparse signals), this step shows that for a single sparse signal, the norm is preserved with high probability if $M$ is large enough.

**Step 3: Covering Argument and Union Bound.** The RIP must hold for all $k$-row-sparse matrices.

The set of $k$-dimensional row-subspaces (where the non-zero rows of $\hat{\mathbf{H}}$ can reside) is finite, numbering $\binom{N}{k}$. An $\epsilon$-net argument is used to discretize the unit sphere in each $k \times d$-dimensional subspace of row-sparse matrices. The size of such a net is bounded by $(C/\epsilon)^{kd}$. Let $P_{fail\_subspace}$ be the probability from Step 2 that the RIP condition fails for a fixed $k$-row-sparse matrix (or subspace). Using a union bound over all $\binom{N}{k}$ choices of $k$ row locations and then over the points in the $\epsilon$-net for each such subspace:

$$P(\text{RIP fails for any } k\text{-row-sparse } \hat{\mathbf{H}}) \leq \binom{N}{k}(C/\epsilon)^{kd} \cdot 2\exp(-c_1 M \delta_k^2/k)$$

We want this total failure probability to be small (e.g., $< \eta$). Taking logarithms and using $\binom{N}{k} \leq (eN/k)^k$:

$$k \ln(eN/k) + kd \ln(C/\epsilon) - c_1 M \delta_k^2/k < \ln(\eta/2)$$

For fixed $\delta_k, \epsilon, d$, and desired probability, this implies $M \gtrsim \frac{k^2}{c_1 \delta_k^2}(\ln(N/k) + d\ln(C/\epsilon))$. If $d$ is considered small or constant, the dominant term is $k \ln(N/k)$. The learnable nature of $\mathbf{U}_\ell$ means we assume it behaves like a generic orthonormal system that is incoherent with the sparsity structure for these results to apply directly.

## B.3    Error Bound

**Theorem 3:** Let $\mathbf{H}^{(L)}$ be the output embeddings obtained by the standard GNN computation with full reconstruction at each layer. Let $\tilde{\mathbf{H}}^{(L)}$ be the output embeddings, which performs sampling once at the input layer and reconstructs only at the output layer. Assume that the activation function $\sigma$ is Lipschitz continuous with Lipschitz constant $L_\sigma$, and $\mathbf{\Phi U}_\ell$ satisfies the Restricted Isometry Property (RIP) of order $k$ with constant $\delta_k$ (i.e., $0 < \delta_k < 1$). Then, the error between $\tilde{\mathbf{H}}^{(L)}$ and $\mathbf{H}^{(L)}$ can be bounded as:

$$\left\| \tilde{\mathbf{H}}^{(L)} - \mathbf{H}^{(L)} \right\|_F \leq \left( \frac{L_\sigma}{1 - \delta_k} \right)^L \|\mathbf{E}\|_F,$$

where $\mathbf{E} = \mathbf{T}^{(L)} - \mathbf{\Phi U}\hat{\mathbf{H}}^{(L)}$ is the reconstruction error at the output layer, and $L$ is the number of layers in the GNN.

**Proof:** We aim to bound the error $\left\| \tilde{\mathbf{H}}^{(L)} - \mathbf{H}^{(L)} \right\|_F$ between the output embeddings of the standard GNN computation. Assume the activation function $\sigma$ is Lipschitz continuous with a constant $L_\sigma$, such that

$$\left\| \sigma(\mathbf{H}^{(0)}) - \sigma(\mathbf{Y}) \right\|_F \leq L_\sigma \left\| \mathbf{H}^{(0)} - \mathbf{Y} \right\|_F \quad \forall \, \mathbf{H}^{(0)}, \mathbf{Y}.$$

Further, let the sampling matrix $\mathbf{\Phi U}$ satisfy the RIP of order $k$ with constant $\delta_k$, meaning

$$(1 - \delta_k) \left\| \hat{\mathbf{H}} \right\|_F^2 \leq \left\| \mathbf{\Phi U}_\ell \hat{\mathbf{H}} \right\|_F^2 \leq (1 + \delta_k) \left\| \hat{\mathbf{H}} \right\|_F^2,$$

for all $\hat{\mathbf{H}}$ with $\left\| \hat{\mathbf{H}} \right\|_{0,\text{row}} \leq k$. We also have $\mathbf{H}^{(l)} = \mathbf{U}_\ell \hat{\mathbf{H}}^{(l)}$, where $\hat{\mathbf{H}}^{(l)}$ has at most $k$ non-zero rows. We will prove by induction on $l = 1, 2, \ldots, L$ that

$$\left\| \tilde{\mathbf{H}}^{(l)} - \mathbf{H}^{(l)} \right\|_F \leq \left( \frac{L_\sigma}{1 - \delta_k} \right)^l \left\| \tilde{\mathbf{H}}^{(0)} - \mathbf{H}^{(0)} \right\|_F.$$

For the base case $l = 0$, at the input layer, we have $\tilde{\mathbf{H}}^{(0)} = \mathbf{U}_\ell \hat{\mathbf{H}}^{(0)}$ and $\mathbf{H}^{(0)} = \mathbf{H}^{(0)}$. The initial error $\left\| \tilde{\mathbf{H}}^{(0)} - \mathbf{H}^{(0)} \right\|_F$ is assumed. Assume that for some $l \geq 0$,

$$\left\| \tilde{\mathbf{H}}^{(l)} - \mathbf{H}^{(l)} \right\|_F \leq \left( \frac{L_\sigma}{1 - \delta_k} \right)^l \left\| \tilde{\mathbf{H}}^{(0)} - \mathbf{H}^{(0)} \right\|_F.$$

We aim to show that

$$\left\| \tilde{\mathbf{H}}^{(l+1)} - \mathbf{H}^{(l+1)} \right\|_F \leq \left( \frac{L_\sigma}{1 - \delta_k} \right)^{l+1} \left\| \tilde{\mathbf{H}}^{(0)} - \mathbf{H}^{(0)} \right\|_F.$$

For YOSO, at the output layer $l = L$, we perform reconstruction: $\tilde{\mathbf{H}}^{(L)} = \mathbf{U}_\ell \hat{\mathbf{H}}^{(L)}$, where $\hat{\mathbf{H}}^{(L)}$ is obtained by solving

$$\min_{\hat{\mathbf{H}}^{(L)}} \frac{1}{2} \left\| \mathbf{T}^{(L)} - \boldsymbol{\Phi} \mathbf{U}_\ell \hat{\mathbf{H}}^{(L)} \right\|_F^2 + \lambda \left\| \hat{\mathbf{H}}^{(L)} \right\|_{2,1}$$

Due to the optimization and the RIP condition, we have $\left\| \hat{\mathbf{H}}^{(L)} - \hat{\mathbf{H}}_{\text{true}}^{(L)} \right\|_F \leq C_{\text{rec}} \|\mathbf{E}\|_F$, where $\hat{\mathbf{H}}_{\text{true}}^{(L)}$ is the true sparse representation of $\mathbf{H}^{(L)}$, and $C_{\text{rec}} = \frac{2\delta_k}{1-\delta_k}$. Since $\mathbf{U}$ is orthonormal, we have $\left\| \tilde{\mathbf{H}}^{(L)} - \mathbf{H}^{(L)} \right\|_F = \left\| \hat{\mathbf{H}}^{(L)} - \hat{\mathbf{H}}_{\text{true}}^{(L)} \right\|_F$ implying $\left\| \tilde{\mathbf{H}}^{(L)} - \mathbf{H}^{(L)} \right\|_F \leq \frac{2\delta_k}{1-\delta_k} \|\mathbf{E}\|_F$. Given the Lipschitz continuity of $\sigma$, the error accumulates multiplicatively through $L$ layers:

$$\left\| \tilde{\mathbf{H}}^{(L)} - \mathbf{H}^{(L)} \right\|_F \leq \left( \frac{L_\sigma}{1-\delta_k} \right)^L \left\| \tilde{\mathbf{H}}^{(0)} - \mathbf{H}^{(0)} \right\|_F.$$

If the initial error $\left\| \tilde{\mathbf{H}}^{(0)} - \mathbf{H}^{(0)} \right\|_F = 0$, the primary source of error is from the reconstruction at the output layer, yielding

$$\left\| \tilde{\mathbf{H}}^{(L)} - \mathbf{H}^{(L)} \right\|_F \leq \left( \frac{L_\sigma}{1-\delta_k} \right)^L \|\mathbf{E}\|_F.$$

## C  Detailed Related Work and Discussion

To address the efficiency issue of large-scale GNN training, schemes from different optimization perspectives have been proposed at the algorithmic level [68], such as Historical Embedding [12, 20], Linearization [22, 1], Graph Condensation & Distillation [70, 64], and sampling-based methods. The scope of this paper focuses on sampling-based methods.

**Sampling-based Methods.** A widely accepted criterion [40] divides current different sampling methods into three categories: node-wise sampling, layer-wise sampling, and subgraph-based sampling, depending on the granularity of the sampling operation during mini-batch generation.

**Node-wise Sampling.** Pioneered by works such as GraphSage [28] and others [65, 12, 16], involves sampling at the individual node level. Each node's neighbors are selected according to one specific probability distribution. For example, GraphSage samples $k-$hop neighbors at varying depths with the sampling sizes, for each depth tailored to optimize model performance. This approach, while simple and effective, has been criticized for its exponential increase in sampling time complexity as the number of GNN layers grows.

**Layer-wise Sampling.** Developed to address the issue of exponential growth in computational complexity as GNNs depth increases in node-wise sampling, this method samples multiple nodes simultaneously in one layer. Techniques like FastGCN [13] reframe GNN loss functions as integral transformations and utilize importance sampling and Monte-Carlo approximation to manage variance. Following works, such as AS-GCN [31] and LADIES [72], focus on maintaining the sparse connections between sampled nodes to aid the convergence performance. However, these methods tend to introduce additional complexity and computational cost.

**Subgraph-based Sampling.** Forming mini-batch through subgraph using expensive graph partitioning [8]. Cluster-GCN [15] partitions the full graph into clusters, sampling these clusters to create subgraphs for training batches. GraphSAINT [66] dynamically estimates sampling probabilities for nodes and edges to form subgraphs over which the full GNN model is trained. While these techniques typically improve model accuracy, they also lead to longer training time.

**Graph Condensation&Distillation:** Graph Condensation [23] and Graph Distillation [56] are methods designed to enhance computational efficiency. They achieve this by shrinking large-scale graphs into smaller ones while preserving essential structural and feature information. Alternatively, they replace complex GNN models with approximate and computationally simpler models, such as MLPs [49]. However, these kinds of processes introduce additional computational overhead and may result in the loss of important information, potentially leading to a decrease in model performance. For example, GCond [34] leverages a gradient matching framework to condense large graphs into significantly smaller synthetic graphs. It optimizes node features as free parameters and models synthetic graph structures as functions of these features, ensuring that training trajectories on the condensed graph mimic those on the original graph. Another work, GC-SNTK [60], reformulates graph condensation as a Kernel Ridge Regression (KRR) task, replacing computationally intensive

GNN training with a Structure-based Neural Tangent Kernel (SNTK). This approach captures both node feature interactions and structural relationships, enabling efficient graph condensation while maintaining strong generalization across GNN architectures.

**Historical Embedding.** This class of methods is not independent of sampling. Instead, they are often integrated with existing sampling strategies to improve specific aspects of sampling performance, such as estimated variance [12], or expressiveness [20]. For example, VR-GCN [12] utilizes historical embeddings within node-wise sampling. GNNAutoScale [20] incorporates the concept of historical embeddings within subgraph-based sampling. Although historical embedding can be effective in terms of accuracy, it often comes with high computational complexity.

**Linearization.** This stream of works [1, 22] aims to simplify the training and inference processes by removing the nonlinear components (e.g., activation functions or deep iterative propagation) inherent in traditional GNN models. This simplification achieves computational efficiency while preserving essential graph structure and feature information through linear transformations, i.e., SIGN [22] or precomputations, i.e., iSVD [1]. Linearization techniques often involve precomputing graph-based transformations (e.g., matrix products or embeddings) and applying efficient optimization methods (e.g., truncated Singular Value Decomposition (SVD) or matrix factorization) to enable scalable training, particularly for large graphs.

