# OpenReview forum: "You Only Spectralize Once: Taking a Spectral Detour to Accelerate Graph Neural Network"
_NeurIPS.cc/2025/Conference — NeurIPS 2025 poster_

### Official Review · Reviewer_zwBn · 2025-06-30

**Clarity:** 2
**Significance:** 2
**Originality:** 3
**Rating:** 3
**Confidence:** 3

**Summary:**

This paper introduces YOSO (You Only Spectralize Once), a novel training scheme for GNNs that leverages spectral sparsity and compressed sensing to accelerate training while maintaining competitive accuracy. The key insight is that node features/embeddings in real-world graphs exhibit sparse representations in the Graph Fourier domain. YOSO projects input features onto a learnable orthonormal Graph Fourier basis Uℓ only once, retaining only M≪N spectral coefficients via a fixed sensing matrix Φ. All GNN computations occur in this compressed M-dimensional subspace. At the output layer, full-graph embeddings are reconstructed.

**Questions:**

- In line 156, unknown equation indicator.
- The authors should elaborate "Uℓ can be initialized as a random orthogonal matrix (e.g., via QR decomposition of a Gaussian matrix) or using other Stiefel manifold initialization techniques".
- The construction of Uℓ still needs matrix decomposition like QR decomposition? Please explain why this is a better choice over eigendecomposition.
- Around line 242, the element-wise product is better denoted by \odot.
- The statement "GNNs iteratively apply the non-linear activation functions. If transform embeddings into spectral domain and later convert them back to original domain (for performing the downstream task), but only use a simple inverse transform (e.g., U^−1), thus, these nonlinearities prevent perfect recovery of original embeddings." seems problematic. Please explain why nonlinearities prevent perfect recovery?


[1] Specformer: Spectral Graph Neural Networks Meet Transformers
[2] A New Perspective on the Effects of Spectrum in Graph Neural Networks

**Ethical Concerns:**

["NO or VERY MINOR ethics concerns only"]

**Final Justification:**

Although the proposed approach aims to speed up GNN training, it still relies on $O(n^3)$ matrix decomposition in preprocessing, which is the main restriction of computational efficiency rather than the layer-wise GNN computations in training. The so-called "You Only Spectralize Once" is misleading, given that existing approaches, like Specformer, etc, also do spectral decomposition once in preprocessing. Also, the proposed learnable basis is also extensively studied by existing work. The authors need to properly contextualize their work with existing work.

**Limitations:**

Yes

**Quality:**

3

**Strengths And Weaknesses:**

#### Strengths

- The integration of learnable spectral transformations with compressed sensing for GNN acceleration is novel. Unlike prior sampling/pruning methods, YOSO avoids repeated spatial operations by operating entirely in a compressed spectral domain after a one-time projection.
- Demonstrates significant speedups while preserving accuracy across diverse tasks (node/link classification) and datasets. Ablation studies confirm robustness w.r.t. M.

#### Weaknesses

- Existing GNNs that require explicit (truncated) eigendecomposition also do eigendecomposition only once at the data preprocessing step, e.g. Specformer [1] and Spec-GN [2]. Given so, the authors need to clarify their main argument "You Only Spectralize Once".
- Experiments focus on GCN. Extensions to other SOTA GNN architectures is not validated.
- The spectral sparsity assumption may not hold for all graph types. The authors are encouraged to provide empirical or theoretical evidence of spectral sparsity assumption.
- The Universal Sensing Matrix $\Phi\in\mathbb R^{M\times N}$ is essential to the main goal of this paper, i.e., compression. But the construction of $\Phi$ in Section 4.3 is not well motivated.

---

> ### Author Rebuttal · Authors · 2025-07-29
>
> Thank you for valuable comments. Please find our response to the listed weaknesses and questions in order.
> ******
> ### Weakness 1：Clarification on the “You Only Spectralize Once”
> We thank the reviewer for pointing out the need to clarify our central message. While it is true that prior works like Specformer and Spec-GN also perform spectralization (i.e., eigendecomposition) once during preprocessing, our use of “You Only Spectralize Once” (YOSO) differs in two key ways:
>
> 1. Avoiding Truncated or Fixed Spectral Bases: YOSO does not rely on a precomputed, truncated eigenbasis. Instead, it employs a learnable orthonormal basis $\mathbf{U}_{l}$ that is jointly optimized with the GNN during training. This addresses the limitations of fixed eigenvectors, which may not align well with the learning objective or generalize across layers.
> 2. One-Time Spectral Projection with Compact Processing: YOSO’s pipeline includes a single spectral projection at the input layer and a single reconstruction at the output layer: entire GNN processing happens in a compressed domain, eliminating repeated transformations or spectral filtering at intermediate layers. In contrast, works like Spec-GN still operate in the spectral domain layer-wise, incurring higher computational complexity during training.
>
> We will clarify this distinction in the final version and revise the wording to emphasize our contribution as a learnable one-time projection and reconstruction framework, as opposed to static preprocessing-based spectralization.
> ******
> ### Weakness 2：Limited evaluation beyond GCN
> YOSO can be applied to any types of transductive GNN architecture like GAT.
>
> The reason is that YOSO's core mechanism is decoupled from the GNN's specific message-passing function. The YOSO pipeline operates as follows:
> 1. Pre-Processing: At the input layer, YOSO takes the full $N\times d$ feature matrix and projects it into a compressed $M\times d$ representation, where $M\ll N$.
> 2. GNN Computation: The GNN model then performs all its layer-wise computations (e.g., aggregation, non-linearities) entirely within this compact $M$-dimensional space.
> 3. Post-Processing: At the output layer, YOSO reconstructs the full $N$-dimensional embeddings from the GNN's final M-dimensional output.
>
> This wrapper approach means that the specific architecture of the GNN does not matter. The GNN simply needs to operate on an input tensor of size $M\times d_{in}$ and produce an output of size $M\times d_{out}$. This is compatible with virtually any transductive architecture.
> *******
> ### Weakness 3: Assumption of graph sparsity
> YOSO's design does not depend on pre-known spectral sparsity of graphs. Instead, our framework is designed to learn an optimal basis that actively creates a sparse representation for the node features (Please also refer to our response to Reviewer tUDz's Question 1, Reviewer mhBm's Weakness 1 and Question 1).
>
> The followings are the theoretical and empirical evidence supporting our claim:
> - Theoretical Justification. The core of our method is a joint optimization objective (Eq.(6)) that includes an $l_{2,1}$-norm regularizer on the spectral coefficients. This regularizer explicitly penalizes non-sparse solutions, forcing the model to learn a basis $\mathbf{U}_{l}$, that is uniquely tailored to sparsify the features of the given dataset. This makes our framework adaptive and robust, even on graph types that do not have an inherently sparse spectrum.
> - Empirical Evidence. In our ablation study, particularly Figure 6. This figure shows the high-fidelity reconstruction of node embeddings from a highly compressed representation ($M\ll N$). According to compressed sensing theory, such an accurate recovery is only possible if the node embeddings were successfully transformed into a sparse-enough representation by our learned basis.
> *****
> ### Weakness 4: Motivation of the construction of $\mathbf{\Phi}$
> Thank you for your comments on the motivation description
>
> The central challenge is to create a single, fixed sensing matrix $\mathbf{\Phi}$ that works robustly with a spectral basis that is learned and evolves during training. Since we cannot know the optimal basis ahead of time, we need a universal sensor. Our solution is a hybrid construction: $\mathbf{\Phi}=\mathbf{S} _ {struct}\otimes \mathbf{\Sigma}_{rand}$, which is designed to satisfy two crucial and complementary objectives:
> 1. Theoretical Guarantees via Randomness ($\mathbf{\Sigma} _ {rand}$): This random component ensures that the reconstruction is theoretically sound. Compressed sensing theory proves that random matrices are excellent universal sensors because they are incoherent with any sparse basis with high probability. This randomness is essential for satisfying the Restricted Isometry Property (RIP), which guarantees that we can faithfully recover the signal regardless of the specific basis $\mathbf{U}_{l}$ the model ultimately learns.
> 2. Graph-Aware Efficiency via Structure ($\mathbf{S}_{struct}$): A purely random approach can be inefficient. To capture the most important information with a smaller number of measurements ($M$), this structural component provides graph-awareness. It guides the compression process by sampling spectral modes based on importance scores derived from the graph Laplacian's eigenvalues. This ensures that high-energy, information-rich spectral components are more likely to be preserved.
>
> In summary, this two-part design creates a single, fixed sensing matrix that is both provably robust (from randomness) and structurally informed (from graph-aware sampling). This hybrid approach is what enables efficient and accurate reconstruction, which is central to YOSO's success. We will revise Section 4.3 in the final version to make this motivation explicit.
> ******
> ### Question 1 and Question 4
> Thank you for pointing them out. The reference on line 156 should be referred to Eq.(1). We will correct the equation reference and notation in the final version.
> ******
> ### Question 2: Initialization of $\mathbf{U}_{l}$
> We thank the reviewer for requesting further elaboration on the initialization of the learnable spectral basis $\mathbf{U}_{l}$.
>
> Because $\mathbf{U} _ {l}$ is constrained to be an orthonormal matrix (i.e., $\mathbf{U} _ {l}^{\top}\mathbf{U}=\mathbf{I}=\mathbf{U}\mathbf{U} _ {l}^{\top}$), its parameters must be initialized on the Stiefel manifold. In our work, we use a simple and standard procedure to achieve this: We first generate a random matrix $\mathbf{A}\in \mathbb{R}^{N \times N}$ by sampling its entries from a standard normal distribution. We then perform a QR decomposition to get $\mathbf{A} = \mathbf{QR}$, where $\mathbf{Q}$ is an orthonormal matrix. We initialize our learnable basis by setting $\mathbf{U}_{l} = \mathbf{Q}$. This is a well-established technique for uniformly sampling from the space of orthogonal matrices.
>
> We will revise the manuscript to include this clarification for completeness and reproducibility.
> *******
> ### Question 3: QR vs. Eigendecomposition
> Although both QR decomposition and eigendcomposition can be used in $\mathbf{U}_{l}$ construction, QR decomposition provide the following advantages for our YOSO design.
> - Numerical Stability and Accuracy for Linear Systems. The Graph Fourier Transform can be viewed as a linear operation. For solving problems related to linear systems, QR decomposition is well-regarded for its numerical stability and accuracy, particularly when dealing with matrices that may be ill-conditioned [1].
> - Applicability to General Matrices. Eigendecomposition, especially the Spectral Theorem which guarantees an orthonormal basis of eigenvectors, applies only to specific classes of matrices (e.g., symmetric or normal matrices). However, the optimal basis $\mathbf{U}_{l}$ learned by YOSO is not constrained to follow such a pattern, it is a general orthonormal matrix adapted for the task. Therefore, QR decomposition, which can be applied to any matrix to produce an orthonormal basis, is a more suitable and general tool for our initialization purposes [2].
> *******
> ### Question 5: Effect of Nonlinearities on Spectral Recovery
> Thank you for this question regarding the impact of nonlinearities on the recovery of original embeddings from the spectral domain. We are happy to clarify the reasoning behind our statement.
>
> Perfect reconstruction after a forward Graph Fourier Transform, a non‑linear activation, and a simple inverse transform is impossible, because the non‑linearity breaks the algebraic symmetry that guarantees invertibility for purely linear operations.
>
> The Graph Fourier Transform (GFT) $\mathbf {U}^{\top}$ and its inverse $\mathbf{U}$ are linear.  All classic reconstruction guarantees hold only when every step between the forward and inverse transforms remains linear.  The moment a non‑linear activation $\sigma(\cdot)$ intervenes, the transform and the activation cease to commute: $\mathbf{U}^{\top}\sigma(\mathbf{H})\neq \sigma(\mathbf{U}^{\top}\mathbf{H})$.
>
> This inequality means that non-linearities mix information across frequencies in a complex way. If we compress the signal in the spectral domain (by discarding coefficients), this mixed information is irretrievably lost. A simple linear inverse transform ($\mathbf{U}$) cannot undo this non-linear mixing to perfectly recover the original signal. This accumulating information loss is precisely what YOSO avoids by performing all non-linear operations within the compressed domain and only reconstructing once at the very end.
>
> We will revise the manuscript to clarify this point more explicitly and avoid any misleading interpretation.
> *******
> ### Reference
> [1] Higham, Nicholas J. Accuracy and stability of numerical algorithms. Society for industrial and applied mathematics, 2002.
>
> [2] Gander, Walter. "Algorithms for the QR decomposition." Res. Rep 80.02 (1980): 1251-1268.

---

> > ### Comment · Reviewer_zwBn · 2025-08-06
> >
> > Replacing eigendecomposition with QR decomposition does not constitute a significant contribution, as consumptive preprocessing is still required, which makes the approach less efficient on large-scale graphs. Moreover, the use of a learnable basis is not novel either, as it has been extensively studied in existing work.

---

> > > ### Author Response · Authors · 2025-08-06
> > > **Reply to QR/Eigendecomposition and Learnable Basis**
> > >
> > > Dear Reviewer zwBn,
> > >
> > > Thank you for the follow-up. We would like to further clarify the novelty of our proposed YOSO below.
> > > ### 1. Clarification on the Novelty of Spectral Domain Training and the Role of QR vs. Eigendecomposition
> > > We would like to clarify that our primary contribution is not the replacement of eigendecomposition with QR decomposition, but rather the proposal of a novel spectral domain training paradigm for GNNs. This approach leverages learnable spectral bases, enabling more expressive and flexible representations than conventional methods. To the best of our knowledge, no prior work has explored direct spectral-domain training of GNNs, nor investigated appropriate initialization strategies within this framework.
> > >
> > > In this context, both eigendecomposition and QR decomposition can serve as means of initializing the learnable spectral basis $\mathbf{U}_{l}$. While eigendecomposition remains a viable option and already provides speed and accuracy benefits over spatial-domain baselines, our investigation extends further: we analyze how different initialization strategies affect training quality and efficiency, particularly for large-scale graphs.
> > >
> > > From a methodological standpoint, we emphasize that a “good” initialization for $\mathbf{U}_{l}$ should be unbiased and information-preserving, offering the optimizer flexibility to adapt the basis to both the graph data and the downstream task. Unlike eigenvectors, which may encode structural biases and are restricted to symmetric matrices, a random orthonormal basis (obtainable via QR decomposition) is more general and task-agnostic. It is also numerically stable and applicable to arbitrary matrices, making it a robust and scalable choice for spectral training (please see Sec. 4.1 of the paper and our response to Reviewer Question 3).
> > >
> > > Overall, the novelty of YOSO is that YOSO redesigns GNN training around a single learnable spectral projection, operates the entire message-passing pipeline in a compressed sensing-informed domain ($M \ll N$), reconstructs the full graph only once at the output layer, thereby yielding an average 74 % reduction in total training time across five benchmarks with no loss of accuracy (Table 1). The way of the training GNN in spectral domain with compressed data is the major contribution of YOSO.
> > >
> > > ### 2. Clarification on the Novelty of the Learnable Basis in YOSO
> > > We would like to clarify that while the concept of a learnable basis is indeed present in prior work (e.g., dictionary learning or signal processing), the way it is applied in our YOSO framework for GNNs is novel.
> > >
> > > Existing approaches typically focus on learning a basis to enhance representation power in Euclidean or grid-structured data. However, graphs differ fundamentally: they are non-Euclidean, irregular, and lack global coordinate systems. Moreover, graph signals often do not follow specific statistical distributions assumed in classical basis learning methods. This makes the design and integration of a learnable basis within GNNs a distinct and nontrivial problems, especially under the constraints of efficiency and scalability.
> > >
> > > In YOSO, the learnable basis $\mathbf{U}_{l}$ is not introduced for representation learning alone. Its novelty lies in how it is co-optimized with a universal sensing matrix $\mathbf{\Phi}$ under a unified objective (Eq. 6) to solve two core GNN challenges: reducing computational overhead and maintaining accuracy. This basis is:
> > > - Task- and sparsity-aware, allowing us to retain only the top-M coefficients without retraining;
> > > - Not only Topology-sensitive but also embedding (Value)-sensitive, ensuring that critical structural information of the graph is preserved, at the same time, maintain the most significant components in embedding;
> > > - Designed to ensure the Restricted Isometry Property (RIP) for the product $\mathbf{\Phi}\mathbf{U}_{l}$, which is essential for stable and accurate recovery of graph signals;
> > > - Central to achieving once-only spectralization, which eliminates redundant per-layer transformations, which is a key to YOSO's efficiency.
> > >
> > > Thus, the novelty is not in the learnability itself, but in how the learnable basis is integrated with other components to enable a new, efficient GNN training paradigm. This synergy is what allows YOSO to overcome longstanding obstacles in spectral GNNs (as discussed in Section 3 and Obstacle I & II), and, to our knowledge, has not been demonstrated in prior work.

---

> ### Author Response · Authors · 2025-08-06
>
> Dear Reviewer zwBn,
>
> We hope this message finds you well.
>
> As the discussion period is drawing to a close with around three days remaining, we wanted to ensure that we have addressed all your concerns satisfactorily. If there are any additional comments or feedback you would like us to consider, please don’t hesitate to let us know; we would be glad to address any concerns you may have. Your insights are invaluable, and we are eager to resolve any remaining issues to further improve our work.
>
> Thank you again for your time and effort in reviewing our paper.

---

### Official Review · Reviewer_mhBm · 2025-07-05

**Clarity:** 3
**Significance:** 4
**Originality:** 3
**Rating:** 5
**Confidence:** 4

**Summary:**

This paper introduces YOSO (You Only Spectralize Once), a novel and practical approach to accelerating GNN training by avoiding repeated message passing. The core idea is to project graph signals into a learned spectral basis and use compressed sensing techniques to sparsely sample and reconstruct signals for downstream tasks like node classification and link prediction. The approach provides strong theoretical guarantees (RIP, error bounds), achieves \~74% training time reduction, and matches or slightly underperforms baselines in accuracy across five benchmark datasets.

**Questions:**

1. How can a practitioner assess whether their dataset satisfies the spectral sparsity assumption needed for successful reconstruction? Are there any heuristics or automatic checks?
2. Given that \$\mathbf{U}\_l\$ is learned during training, how do you ensure that the RIP condition is preserved or approximately satisfied?
3. Have you considered whether the learned spectral basis can adapt incrementally to graph topology changes, or would retraining be required?

**Ethical Concerns:**

["NO or VERY MINOR ethics concerns only"]

**Limitations:**

1. The method assumes spectral sparsity in the node feature signals, which may not generalize well to graphs with dense or noisy spectral representations (e.g., social networks, small-world graphs). The paper would benefit from practical guidance or automatic checks for this assumption before deployment.

2. While the method excels on static graphs, its extension to dynamic or evolving graphs remains unclear. If the graph topology changes over time, the learned spectral basis may no longer be valid, potentially requiring full retraining. This could limit real-world applicability in streaming or online settings.

3. The current evaluation is restricted to transductive learning. It is not evident how the framework would perform on inductive node classification or in few-shot graph scenarios, where node embeddings need to generalize to unseen subgraphs.

4. The reconstruction module (decoder) plays a crucial role in performance, but its architecture and training stability are underexplored. In particular, the decoder’s ability to recover clean signals across different graph structures is not analyzed in depth.

5. The approach requires the initial spectralization step to be performed once, but this may still involve expensive eigen decomposition or approximation for large graphs, depending on the implementation. Further analysis on this one-time cost would improve transparency.

**Quality:**

4

**Strengths And Weaknesses:**

### **Strengths:**

#### **Technical Quality:**

* Provides rigorous theoretical backing for the reconstruction process, including proofs for the sensing matrix’s full-rank property and RIP satisfaction.
* The integration of learnable GFTs with compressed sensing is implemented carefully and supported by mathematical justification and empirical results.
* Training speed improvements are consistent and significant across tasks and datasets.

#### **Significance:**

* The work directly addresses a major bottleneck in large-scale GNN training — computational inefficiency — with a conceptually novel and practical solution.
* Proposes a new paradigm shift from spatial-domain sampling (used in methods like GraphSAINT or Cluster-GCN) to spectral-domain sparse sampling.

#### **Originality:**

* While compressed sensing and GFTs are established fields, their combination with learnable spectral bases in GNNs is novel.
* The “you only spectralize once” conceptually reframes how feature propagation is handled — offering a fresh direction for GNN efficiency research.


* The paper is well-organized and the core idea is clearly presented.
* Appendices provide implementation details and proofs, supporting reproducibility.

---

### **Weaknesses:**

* The method relies on spectral sparsity, which may not hold in all graph types (e.g., complete graphs, random graphs). More guidance or diagnostics for real-world practitioners would help.
* There is limited discussion on dynamic or time-evolving graph settings, where spectral bases may evolve.
* Ablation experiments isolating the contributions of GFT learning vs. sampling vs. decoder reconstruction could strengthen the empirical section.
* A minor typo: Equation reference on line 156 is not rendered correctly in the PDF.

---

> ### Author Rebuttal · Authors · 2025-07-29
>
> Thank you for the insightful review. We address the weaknesses, questions and the limitations as below.
>
> ***
> ### Weakness 1: On the Assumption of Sparsity for Graphs and Practical Guidance
> As our response to Reviewer tUDz's Question 1, YOSO does not rely on pre-known spectral sparsity. This is a key distinction that enables YOSO applied any type of static graphs. On the contrary, YOSO learns an optimal orthonormal basis $\mathbf{U}_l$ that actively constructs a sparse representation of the graph features. Learning process is entirely data-driven and does not pre-suppose any inherent sparsity in the input graphs. Therefore, even on a graph with a dense spectrum (like a complete or random graph), if node features themselves have a compressible structure (e.g., nodes belong to a few classes), YOSO is designed to learn a basis that captures this structure sparsely.
>
> Guidance for Practitioners: People do not need to pre-diagnose graph for spectral sparsity due to the above discussion. YOSO is designed to automatically discover compressible data structures during training. Its applicability is broad, precisely because it creates an efficient compressed domain rather than assuming one exists beforehand.
>
> ***
> ### Weakness 2: On Applicability to Dynamic Graphs
> YOSO is possible to accelerate training for dynamic graphs, depending on how the graph evolves over time (Please also refer to Reviewer tUDz's Question 2).
>
> YOSO can achieve shorter training time for discrete-time (snapshot-based) dynamic graphs [1,2,3,4], which covers a wide range of real-world scenarios [5]. YOSO can be applied independently to each snapshot, which accelerates the training process.
>
> While for continuous-time (event-based) graphs [6,7], YOSO may not be able to obtain the training time reduction since every change in the graph requires re-computation of global basis.
>
> Thanks for raising this question, and we will add the discussion to the limitation section.
>
> ***
> ### Weakness 3: On an Ablation Study of YOSO's Components
> We appreciate the reviewer’s suggestion that ablation experiments isolating the contributions of GFT learning ($\mathbf{U}_{l}$), sampling (sensing $\mathbf{\Phi}$), and decoder (reconstruction process) could strengthen the empirical section. However, in our current formulation, these three components are tightly coupled, and their contributions cannot be meaningfully disentangled without fundamentally altering the method, as explained below:
> - GFT learning ($\mathbf{U}_l$) is driven by the reconstruction loss, which is computed by the decoder. The gradients from this loss directly update $\mathbf{U}_l$, meaning the learned basis is specifically optimized to facilitate faithful reconstruction by the decoder. Without the decoder, the GFT learning objective becomes ill-defined.
> - Decoder (reconstruction, using our FISTA-based solver) solves the inverse problem induced by the compressed measurements produced by the sensing matrix $\mathbf{\Phi}$. Its performance is intrinsically tied to how $\mathbf{\Phi}$ samples the sparse representation, and thus cannot be evaluated independently.
> - The GFT $\mathbf{U}_{l}$ and sensing matrix $\mathbf{\Phi}$ are dependent with each other. The GFT learns to make the signal sparse, which is the necessary precondition for the compressed sensing via $\mathbf{\Phi}$ to be effective. One component creates the condition (sparsity) that the other is designed to exploit.
>
> ***
> ### Weakness 4: Typo
> The reference on line 156 should point to Eq.(1). We'll fix this.
>
> ***
> ### Question 1: Assessing Sparsity and Reconstruction of Datasets
> Practitioners do not need to perform any pre-assessment of their datasets/graphs for spectral sparsity. Our method is designed to be broadly applicable without such checks. The reason is that YOSO does not rely on the input graph having an inherently sparse structure. Instead, it learns an optimal orthonormal basis ($\mathbf{U}_{l}$) that creates a sparse representation of the node features for the specific task at hand (Please also refer to our response to Reviewer tUDz's Question 1 and the response to Weakness 1).
>
> Successful reconstruction is guaranteed by satisfying the Restricted Isometry Property (RIP), which is a core part of YOSO's design, as explained below:
> - The GFT learning process, guided by the $l_{2,1}$ regularization in our loss function, explicitly trains $\mathbf{U}_{l}$ to find a basis where the node features can be represented sparsely.
> - Our specialized sensing matrix ($\mathbf{\Phi}$), which combines graph-aware structure and randomness, is designed to work in concert with the learned $\mathbf{U}_{l}$. As established in our Theorem 2, this joint design ensures that the RIP condition is satisfied with high probability.
>
> ***
> ### Question 2: Preserving the RIP Condition with a Learned Basis
> RIP is satisfied by co-adapting the learnable basis with the fixed sensing matrix. This co-adaptation is achieved through our proposed joint optimization objective:
> - The sensing matrix $\mathbf{\Phi}$ is constructed once with a random component. This design ensures it has good RIP-like properties for generic orthonormal bases, making it a robust all-purpose sensor.
> - The basis  $\mathbf{U} _ {l}$ does not evolve randomly and it is updated via gradients from the total loss $\mathcal{L} _ {\text{total}}$. Two key components of this loss guide its evolution: (a) The sparsity regularizer ($l_{2,1}$-norm) pushes $\mathbf{U} _ {l}$ to become a basis that makes the signal sparse; (b) The data consistency term ($||\mathbf{T} ^ {(L)} - \mathbf{\Phi}\mathbf{U} _ {l}\hat{\mathbf{H}} _ {c} ^ {(L)*}|| _ {F}^{2}$) penalizes any changes to $\mathbf{U}_{l}$ that make the signal difficult to reconstruct from the measurements taken by the fixed $\mathbf{\Phi}$.
>
> ***
> ### Question 3: Incremental Adaptation vs. Retraining for Topology Changes
> Please refer to the response to Weakness 2.
>
> ***
> ### Limitation 1: Assumption of spectral sparsity in node features
> Please refer to the response to Weakness 1 and Question 1.
>
> ***
> ### Limitation 2: Applicability to dynamic or evolving graphs
> Please refer to the response to Weakness 2.
>
> ***
> ### Limitation 3: Generalization to inductive and few-shot scenarios
> Thank you for raising this point. YOSO is currently transductive. Both $\mathbf{U}_{l}$ and $\mathbf{\Phi}$ depend on the full node set of size $N$, so they cannot process new nodes at inference.
> Transductive settings still cover many important applications [8–11], which motivated our design.
> Generalizing YOSO to inductive settings is an important future direction, and we will discuss this in the paper.
>
> ***
> ### Limitation 4: Underexplored decoder design and stability
> We're happy to provide more details on the reconstruction module's design and its robustness.
> - Ability of Recover Clean Signals vs. Graph Structure
>
> The decoder's (reconstruction process) ability to recover clean signals is not directly dependent on the graph's structure but on a more fundamental condition: whether the Restricted Isometry Property (RIP) is satisfied. Our framework is designed to ensure this holds true with high probability regardless of the graph topology. We have formally proven this with Theorem 2 in Appendix C.2.
> Besides, we also provide direct empirical evidence in Figure 6. The heatmap visualizes the near-zero absolute difference between original and reconstructed embeddings on the ogbn-products dataset, confirming the decoder's ability to achieve high-fidelity recovery in practice.
> - Decoder Architecture and Stability
>
> It's important to clarify that our decoder is not a learnable function but an implementation of the FISTA algorithm [7], a well-established and provably convergent iterative solver. This directly addresses the architecture and stability concerns: (a) Architecture: The architecture is a standard optimization algorithm, widely used for solving $l_{2,1}$-regularized problems like the one in our framework; (b) Stability: As FISTA is a stable and convergent algorithm [7], its stability during training is not a primary concern.
> ***
> ### Reference
> [1] Pareja, Aldo, et al. "Evolvegcn: Evolving graph convolutional networks for dynamic graphs." Proceedings of the AAAI conference on artificial intelligence. Vol. 34. No. 04. 2020.
>
> [2] You, Jiaxuan, et al.. "ROLAND: graph learning framework for dynamic graphs." Proceedings of the 28th ACM SIGKDD conference on knowledge discovery and data mining. 2022.
>
> [3] Seo, Youngjoo, et al. "Structured sequence modeling with graph convolutional recurrent networks." International conference on neural information processing. Cham: Springer International Publishing, 2018.
>
> [4] Bonner, Stephen, et al. "Temporal neighbourhood aggregation: Predicting future links in temporal graphs via recurrent variational graph convolutions." 2019 IEEE international conference on big data (Big Data). IEEE, 2019.
>
> [5] Zheng, Yanping, Lu Yi, and Zhewei Wei. "A survey of dynamic graph neural networks." Frontiers of Computer Science 19.6 (2025): 196323.
>
> [6] Trivedi, Rakshit, et al. "Dyrep: Learning representations over dynamic graphs." International conference on learning representations. 2019.
>
> [7] Rossi, Emanuele, et al. "Temporal graph networks for deep learning on dynamic graphs." arXiv preprint arXiv:2006.10637 (2020).
>
> [8] Zhu, Xiaojin, et al. "Semi-supervised learning using gaussian fields and harmonic functions." Proceedings of the 20th International conference on Machine learning (ICML-03). 2003.
>
> [9] Zhang, Yan-Ming, et al. "Transductive learning on adaptive graphs." Proceedings of the AAAI Conference on Artificial Intelligence. Vol. 24. No. 1. 2010.
>
> [10] Wang, Jun, et al. "Active microscopic cellular image annotation by superposable graph transduction with imbalanced labels." 2008 IEEE Conference on Computer Vision and Pattern Recognition. IEEE, 2008.
>
> [11] Valko, Michal, et al. "Online semi-supervised learning on quantized graphs." arXiv preprint arXiv:1203.3522 (2012).

---

### Official Review · Reviewer_tUDz · 2025-07-07

**Clarity:** 3
**Significance:** 3
**Originality:** 3
**Rating:** 5
**Confidence:** 3

**Summary:**

The paper proposes a GNN training framework called You Only Spectralize Once (YOSO). They first project the features into a learnable orthonormal graph Fourier basis, and then use compressed sensing, which retains a subset of the spectral coefficients. Then the GNN computation is performed in the compressed spectral domain. Finally, they recover the full graph embeddings to the original domain at the output layer.

**Questions:**

1. The proposed method relies on the assumption that the graph features are sparse in the spectral basis. I am curious whether this assumption applies to graphs with noises or heterogeneous features. Is there any evidence or analysis to support this assumption?
2. Can this method be adapted to the dynamic graphs? For example, after the matrix $U_l$ is trained, can it be reused or transferred when the graph is changing over time?

**Ethical Concerns:**

["NO or VERY MINOR ethics concerns only"]

**Final Justification:**

I have read the rebuttal and other reviews. I maintain the original score.

**Limitations:**

yes

**Quality:**

4

**Strengths And Weaknesses:**

Strength:
1. The proposed method achieves consistent training time reductions without major accuracy loss. The experiments are extensive and well-controlled.
2. The paper offers a clear and well-motivated explanation of the problem and the solution. The background on spectral graph theory and compressed sensing makes it easier for readers to understand the core concepts.
3. I appreciate Figure 2, which effectively summarizes YOSO.
4. Each step in the framework is well-motivated and thoughtfully designed. For example, the method is grounded in compressed sensing theory and spectral graph signal processing. The RIP-based analysis provides a solid theoretical justification for the design. Also, the idea of using FISTA is a clever choice, effectively integrating traditional compressed sensing techniques with modern learning-based approaches.

Weakness:
See questions.


Minor:
1. Line 17 ‘we prove that stable recovery by….’
2. Line 156 ‘Eq (??)’

---

> ### Author Rebuttal · Authors · 2025-07-29
>
> We are grateful for your thorough review. We address your specific questions below.
> *****
> ### Question 1: Assumption of Sparsity for Graphs
> We would like to clarify that our proposed scheme, YOSO, does not rely on the assumption that graph features are sparse in pre-known spectral basis. This is a key distinction that enables YOSO to be applicable to any type of static graphs, including those with noisy or heterogeneous features.
>
> YOSO is designed to learn an optimal orthonormal basis $\mathbf{U}_{l}$ that actively constructs a sparse representation of the graph features. This learning process is entirely data-driven and does not pre-suppose any inherent sparsity in the input graphs. As a result, YOSO remains effective even when such sparsity assumptions fail to hold.
>
> Theoritical analysis has been provided in the paper to indicate:
> - How the learning process of $\mathbf{U} _ {l}$ is independent of feature sparsity?
>
> As established in Theorem 2, this is achievable because our learned $\mathbf{U} _ {l}$ is regularized and guided to be sufficiently incoherent with the sparsity basis. The learning of $\mathbf{U} _ {l}$ is guided by our joint optimization objective $\mathcal{L} _ {\text{total}}$, which ensures that the sparse representation retains sufficient information for (mostly) lossless reconstruction of the original graph features. A key component of this objective is the reconstruction loss $\mathcal{L} _ {\text{recon}}$, which incorporates an $l _ {2,1}$-norm regularization term: $\lambda || \hat{\mathbf{H}} _ {c} ^ {(L)*} || _ {2,1}$. This regularizer encourages the model to discover a transformation $\mathbf{U} _ {l}$ that maps the input features into a sparse representation by penalizing non-sparse outputs. As a result, the training process explicitly promotes sparsity while preserving the fidelity of the original signal.
> - Why YOSO can recover the sparse graph representation back to original graphs?
>
> The key insight is that successful reconstruction via Compressed Sensing (CS) depends on the Restricted Isometry Property (RIP) of the combined sensing and transformation operator $\mathbf{\Psi} = \mathbf{\Phi}\mathbf{U} _ {l}$. As shown in Theorem 3, if the RIP condition holds, the error between our reconstructed output embeddings ($\tilde{\mathbf{H}} ^ {(L)}$) and the idealized full-graph embeddings $\mathbf{H} ^ {(L)}$ is bounded: $|| \tilde{\mathbf{H}} ^ {(L)} - \mathbf{H} ^ {(L)} ||_ F \leq \frac{L _ {\sigma}}{1-\delta_k}||E|| _ F$. This means the final error is controllably proportional to the CS solver's reconstruction error ($E$), which is directly minimized during training.
> *****
> ### Question 2: On Applicability to Dynamic Graphs
> YOSO could be extended to dynamic graphs, depending on how the graph evolves over time.
>
> For discrete-time dynamic graphs (snapshot-based) [1,2], where the dynamic graph is represented as a sequence of static snapshots, YOSO can be applied independently to each snapshot. This allows us to accelerate GNN training at each time step, making YOSO applicable to many real-world dynamic graph scenarios that process data in temporal batches.
>
> For continuous-time dynamic graphs (event stream-based) [3,4], where graph structure and node features change continuously, applying YOSO may not improve the training efficiency. Since the core matrices $\mathbf{U} _ {l}$ and $\mathbf{\Phi}$ are defined based on the global node set $N$, every change in the graph requires re-computation or adaptation of these matrices, which would introduce large overhead and diminish the computational advantages of YOSO.
>
> We appreciate the reviewer’s insight and will incorporate this discussion into the limitations section of the paper to clarify the scope and potential extensions of our method.
> ******
> ### Minors
> Thank you for catching these typos. We will add the following corrections in the final version:
>
> Line 17: The grammatical error will be corrected to read, "...we prove that stable recovery is possible by showing that this whole process can satisfy the Restricted Isometry Property..."
> Line 156: The reference to Eq.(??) refers to Eq.(1).
>
> *******
> ### Reference
> [1] Pareja, Aldo, et al. "Evolvegcn: Evolving graph convolutional networks for dynamic graphs." Proceedings of the AAAI conference on artificial intelligence. Vol. 34. No. 04. 2020.
>
> [2] You, Jiaxuan, Tianyu Du, and Jure Leskovec. "ROLAND: graph learning framework for dynamic graphs." Proceedings of the 28th ACM SIGKDD conference on knowledge discovery and data mining. 2022.
>
> [3] Trivedi, Rakshit, et al. "Dyrep: Learning representations over dynamic graphs." International conference on learning representations. 2019.
>
> [4] Rossi, Emanuele, et al. "Temporal graph networks for deep learning on dynamic graphs." arXiv preprint arXiv:2006.10637 (2020).

---

> > ### Comment · Reviewer_tUDz · 2025-08-07
> >
> > Thank you for the answer. I will keep the score.

---

### Comment · Area_Chair_3anK · 2025-08-02

Dear Reviewers

The authors have responded to your reviews. In the next few days, please read their responses and engage in a productive discussion that will be critical to the review process.

I truly appreciate your timely thoughts and comments!

AC

---

### Decision · Program_Chairs · 2025-09-17

**Decision:**

Accept (poster)

**Comment:**

After reading the paper, the reviews, and the discussion, I have decided to recommend accepting this submission. One of the main concerns raised by the reviewers was the sparsity assumption, and it seems to be addressed during the rebuttal period. While I agree with the reviewer zwBn that the paper can be further improved, I think that there is enough novelty to warrant acceptance.